# VideoTemp-o3: Harmonizing Temporal Grounding and Video Understanding in Agentic Thinking-with-Videos

**Wenqi Liu** [1][*]   **Yunxiao Wang** [1][*]   **Shijie Ma** [2][*]   **Meng Liu** [1][†]   **Qile Su** [3]
**Tianke Zhang** [4]   **Haonan Fan** [4]   **Changyi Liu** [4]   **Kaiyu Jiang** [4]   **Jiankang Chen** [4]   **Kaiyu Tang** [4]
**Bin Wen** [4][‡]   **Fan Yang** [4]   **Tingting Gao** [4]   **Han Li** [4]   **Yinwei Wei** [1]   **Xuemeng Song** [5][†]

https://liuwq-bit.github.io/VideoTemp-o3

## Abstract

In long-video understanding, conventional uniform frame sampling often fails to capture key visual evidence, leading to degraded performance and increased hallucinations. To address this, recent agentic *thinking-with-videos* paradigms have emerged, adopting a *localize–clip–answer* pipeline in which the model actively identifies relevant video segments, performs dense sampling within those clips, and then produces answers. However, existing methods remain inefficient, suffer from weak localization, and adhere to rigid workflows. To solve these issues, we propose `VideoTemp-o3`, a unified agentic *thinking-with-videos* framework that jointly models video grounding and question answering. `VideoTemp-o3` exhibits strong localization capability, supports on-demand clipping, and can refine inaccurate localizations. Specifically, in the supervised fine-tuning stage, we design a unified masking mechanism that encourages exploration while preventing noise. For reinforcement learning, we introduce dedicated rewards to mitigate reward hacking. Besides, from the data perspective, we develop an effective pipeline to construct high-quality long video grounded QA data, along with a corresponding benchmark for systematic evaluation across various video durations. Experimental results demonstrate that our method achieves remarkable performance on both long video understanding and grounding.

[*]Equal contribution [‡]Project Leader [1]Shandong University [2]Institute of Automation, Chinese Academy of Sciences [3]Beihang University [4]Kuaishou Technology [5]Southern University of Science and Technology. Correspondence to: Meng Liu <mengliu.sdu@gmail.com>, Xuemeng Song <sxmustc@gmail.com>.

*Proceedings of the 43rd International Conference on Machine Learning*, Seoul, South Korea. PMLR 306, 2026. Copyright 2026 by the author(s).

## 1. Introduction

Although Multimodal Large Language Models (MLLMs) (Bai et al., 2025; Team et al., 2025b;a) have made remarkable advances, their internal knowledge remains static once trained, limiting their capacity to tackle complex tasks in dynamic environments. To address this issue, recent works introduce agentic MLLMs capable of invoking external tools. For example, when encountering up-to-date and knowledge-intensive questions, agentic models can proactively invoke search tools (Wu et al., 2025; Narayan et al., 2025). For fine-grained perception (Tong et al., 2024; Ma et al., 2025b) of high-resolution inputs, OpenAI-o3 has pioneered the *thinking-with-images* paradigm (Zheng et al., 2025; Zhang et al., 2025b; Hong et al., 2025), enabling image transformations that amplify salient details while suppressing irrelevant content.

In long-video understanding (Feng et al., 2025; Li et al., 2025b), conventional approaches typically rely on uniform frame sampling under a fixed frame budget to control computational cost. Such a rigid pipeline yields sparse visual evidence and often overlooks query-relevant content, leading to substantial performance degradation. Therefore, taking the spirit of *thinking-with-images*, *thinking-with-videos* has emerged to enable query-aware fine-grained temporal grounding (Liu et al., 2018; Mun et al., 2020; Wang et al., 2022; Lin et al., 2023; Liu et al., 2023; Wang et al., 2026). In this paradigm, MLLMs first proactively identify the video segments most pertinent to the question. They then perform dense sampling within these intervals to produce reliable, clip-grounded responses. In the literature, VideoExplorer (Yuan et al., 2025) proposes a multi-agent framework composed of a planner, a grounder, and an understanding model, which collaboratively achieve grounded video understanding. Later works like VITAL (Zhang et al., 2025a), REVISOR (Li et al., 2025a), and LongVT (Yang et al., 2025) adopt hybrid training that combines Supervised Fine-Tuning (SFT) and Reinforcement Learning (RL) on curated datasets, enabling models to effectively identify key segments and produce accurate responses.

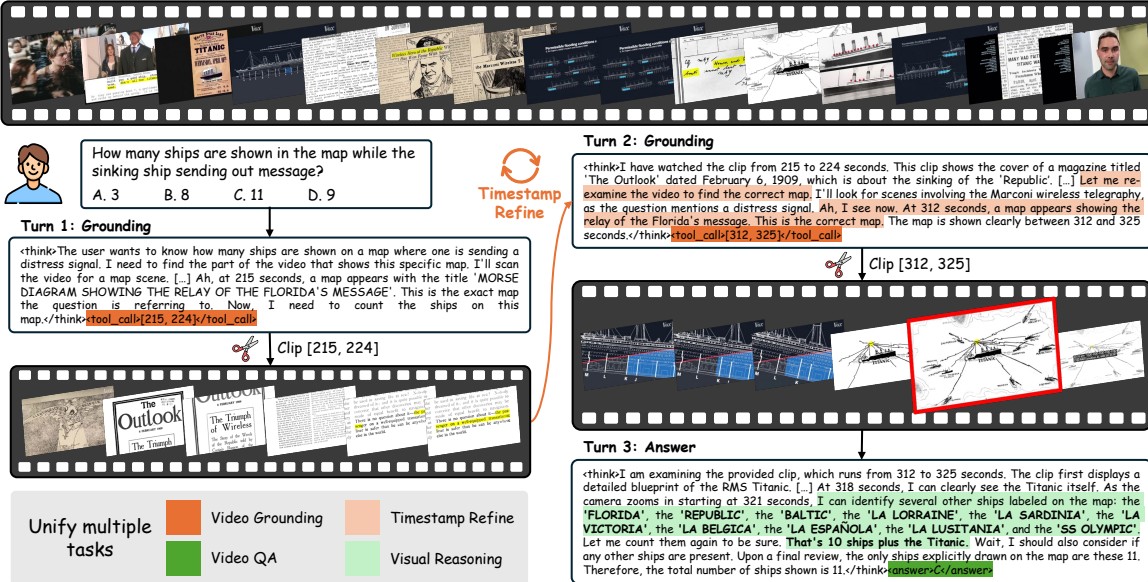

*Figure 1.* Illustration of the agentic pipeline in `VideoTemp-o3`. Given the video QA pair, it performs on-demand grounding and refines the initial rough segment. Finally, it produces a reliable answer grounded in the pertinent visual evidence.

Despite these advancements, several critical limitations persist: (1) *Workflow complexity.* Many existing methods rely on multiple specialized models to separately perform temporal grounding and video question answering, incurring substantial inference overhead. (2) *Imprecise grounding.* Many approaches struggle with precise grounding and offer limited mechanisms to evaluate or refine grounding results. (3) *Rigid pipelines.* Most methods follow a rigid procedure that the model blindly crops the video *once* and then immediately produces an answer, leading to unnecessary computation for short videos that could be answered directly, while lacking support for iterative grounding refinement in challenging long-video scenarios. We attribute these limitations to three primary factors: (1) *Suboptimal training strategies.* Existing SFT supervision and RL reward designs are insufficient to promote precise grounding or support iterative refinement of inaccurate intervals. (2) *Low-quality data.* Current grounding annotations are generally shifted and are dominated by short videos. (3) *Lack of instructional trajectories to incentivize desirable patterns.* There is a lack of high-quality, multi-turn trajectories that exemplify *localize-clip-answer* behavior, hindering MLLMs from effectively internalizing the *thinking-with-videos* patterns.

To address these challenges, we propose `VideoTemp-o3`, a unified *thinking-with-videos* framework that jointly integrates video question answering (VideoQA) and temporal grounding within a single model. `VideoTemp-o3` supports on-demand grounding and cropping, enabling iterative refinement until sufficient evidence is gathered to produce a reliable answer, as in Fig. 1. Specifically, we begin by establishing a foundational *localize-clip-answer* capability

through cold-start SFT. To facilitate learning across both VideoQA and grounding, we introduce a unified masking strategy for multi-turn supervision that encourages exploration while preserving reliable learning signals. In the subsequent RL stage, we design tailored rewards to significantly improve grounding precision and clip-grounded answer accuracy, while effectively mitigating reward hacking. In this way, `VideoTemp-o3` strengthens its internal grounding ability, which is the prerequisite for *thinking-with-videos* paradigm. From the data perspective, we design a dedicated pipeline to construct large-scale long-video grounded QA (GQA) datasets with accurate temporal segments and corresponding answers, ensuring strong alignment between grounded evidence and responses. Finally, we introduce VideoTemp-Bench, a benchmark for evaluating GQA across videos of diverse durations, as well as in-depth analyses.

In summary, our main contributions are as follows:

- We present `VideoTemp-o3`, an agentic *thinking-with-videos* model that harmonizes temporal grounding and VideoQA within a single architecture. It supports on-demand video cropping and enables multi-turn grounding refinement.

- We develop a cold-start SFT strategy and tailor-made RL rewards to enhance the model's internal grounding performance. The training paradigm effectively instills *thinking-with-videos* behavior and yields stronger video understanding.

- We introduce a high-quality pipeline to curate a large-scale long-video GQA dataset. The resulting data tightly aligns cropped segments with answers.

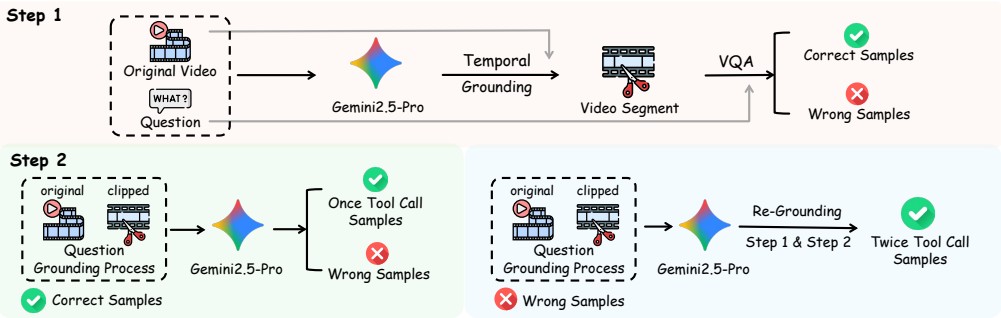

*Figure 2.* Multi-turn, multi-tool call data curation pipeline.

- Extensive experiments demonstrate that `VideoTemp-o3` achieves state-of-the-art performance across several video understanding benchmarks. We also present VideoTemp-Bench that highlights the limitations of current models and provides in-depth analyses.

## 2. Related Work

**Agentic Multimodal Large Language Models.** Agentic MLLMs substantially improve real-world problem-solving by leveraging external tools. In particular, for up-to-date or information-seeking questions beyond the model's internal knowledge, agentic search models (Wu et al., 2025; Narayan et al., 2025) can effectively identify the boundaries of their capabilities. When needed, they autonomously invoke search engines to retrieve external information, enabling more accurate responses and reducing hallucinations. In scenarios requiring fine-grained perception (Tong et al., 2024; Ma et al., 2024; 2025c), such as those involving small foreground objects, OpenAI-o3 introduced the *thinking-with-images* paradigm (Zheng et al., 2025), which enables models to invoke external tools for operations like cropping and zooming to better extract critical visual cues, and significantly enhances fine-grained understanding. Inspired by this, Thyme (Zhang et al., 2025b) and DeepEyesV2 (Hong et al., 2025) leverage code generation as a universal interface for flexible image manipulation. Recently, the *thinking-with-videos* paradigm has been proposed, extending fine-grained perception of *thinking-with-images* from the spatial dimension to the temporal dimension.

**Long Video Understanding.** Recent works like VideoChat-R1 (Li et al., 2025b) and Video-R1 (Feng et al., 2025) have adopted RL-based strategies to improve general video understanding and reasoning. However, long video understanding remains challenging. A key bottleneck is the prevalent use of uniform sampling under a fixed frame budget, introduced to control computational cost, which often misses critical visual evidence. To alleviate this, LongVA (Zhang et al., 2024) and LongVILA (Chen et al., 2024c) employ long-context fine-tuning to extend context length for long videos. Others (Li et al., 2024; Hyun et al., 2025; Wang et al., 2025d) resort to visual token compression. More recently, the agentic *thinking-with-videos* paradigm has emerged as a complementary approach. Some works, including VideoChat-R1.5 (Yan et al., 2026), VideoThinker (Wang et al., 2025b), Open-o3 Video (Meng et al., 2025), and Rewatch-R1 (Zhang et al., 2026), can output timestamps or bounding boxes during reasoning, but they mainly follow a text-only CoT paradigm without explicit tool calls, and therefore provide limited support for multi-round localization and answering. Another line of work first localizes or selects relevant visual content and then answers based on the selected evidence. For example, VideoExplorer (Yuan et al., 2025) leverages multi-agent collaboration, with specialized modules handling planning, grounding, and understanding. VITAL (Zhang et al., 2025a) adopts a two-stage training process involving SFT-RL to elicit the model's segment cropping. REVISOR (Li et al., 2025a) introduces a two-stage inference paradigm that first grounds the relevant intervals and then generates responses based on the selected content. Similarly, LongVT (Yang et al., 2025) proposes a three-stage training SFT-RL-RFT strategy to progressively enhance *thinking-with-videos* behavior. FrameThinker (He et al., 2026) also performs multi-turn visual selection before answering by spotlighting informative frames. These methods demonstrate the effectiveness of focusing on key visual evidence, while our work further unifies temporal grounding and VideoQA in a single model with on-demand cropping, iterative refinement, and dedicated training designs for noisy multi-turn trajectories.

## 3. Dataset and Benchmark

### 3.1. Data Construction Pipeline

Existing datasets exhibit several limitations that hinder the induction of *thinking-with-videos* behavior. (1) Grounding annotations are often low-quality, with coarse and temporally misaligned intervals. (2) Long video QA with explicit temporal grounding annotations is scarce. (3) Since accurate grounding in a single pass is sometimes unrealistic in long

videos, MLLMs must learn reflection, *i.e.*, refining grounding results through multi-turn cropping. To address these limitations, we propose a dedicated pipeline for constructing both single-turn and multi-turn data.

**Single-turn Data without Tool Call.** To establish fundamental reasoning ability, we construct high-quality Chain-of-Thought (CoT) training samples for both QA and grounding. We employ a rejection sampling strategy to ensure reliable reasoning chains. Specifically, for the VideoQA data, we utilize Qwen3-VL-235B-A22B-Thinking (Bai et al., 2025; Team, 2025) to generate both the reasoning trajectories and final answers. Only samples whose predicted answers match the ground truth are retained. For grounding, we re-annotate and filter data to improve temporal precision. Concretely, we leverage Gemini-2.5-Pro (Comanici et al., 2025) to re-localize key segments and keep samples whose predicted intervals achieve IoU>0.5 with the original annotations, ensuring precise and reliable grounding labels for training.

**Multi-turn Data with Tool Call.** The core objective of multi-turn data is to simulate realistic, tool-assisted grounding behaviors. By iteratively identifying, cropping, and validating relevant video segments for a given QA task, the framework enables the construction of high-quality multi-turn training samples that explicitly involve reflection and verification of previously localized content. A key component of our design is a dedicated verification stage, which critically evaluates whether the selected video segments provide sufficient evidence to support a coherent *thinking-with-videos* reasoning chain.

As shown in Fig. 2, in Step 1, given an original video and its question, we first prompt Gemini-2.5-Pro (Comanici et al., 2025) to predict a candidate temporal segment. We crop this segment and feed it into the model along with the original question, while explicitly constraining the model to answer the question solely based on the segment. Only samples whose predicted answers match the ground truth are retained. This filtering step ensures that the localized video segment contains sufficient information to correctly answer the question. In Step 2, we perform a closed-loop consistency verification. Specifically, we provide the model with the full context, including original video, question, grounding process and cropped segment. We then instruct the model to reason based on the localized segment and prior context. Samples are retained only if the final answer remains consistent with the ground-truth, forming multi-turn, single-tool-call training data. For samples that fail either verification stage, we prompt Gemini-2.5-Pro to re-ground the video by leveraging the accumulated context from the previous failure attempt, and then repeat both step 1 and step 2. Samples that pass after one additional refinement round are retained as multi-turn, multi-tool-call training data. To ensure that iterative grounding is truly necessary, we apply this procedure only to long videos (>3min). For

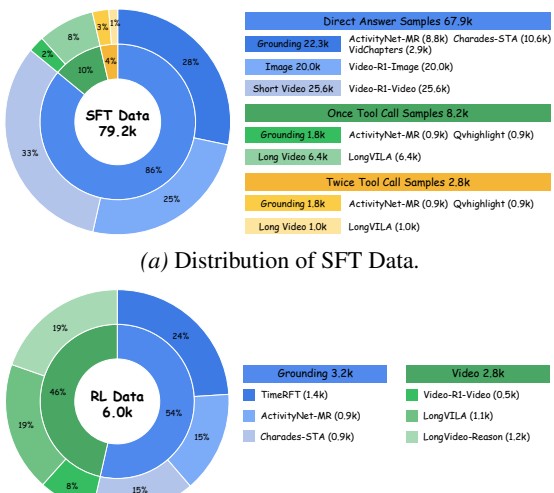

*(a)* Distribution of SFT Data.

*(b)* Distribution of RL Data.

*Figure 3.* Training Data Distribution.

questions with multiple possible evidence segments, this verification process ensures that the selected segment is sufficient to answer the question, even if other valid evidence also exists elsewhere in the video.

### 3.2. Training Data

**SFT Data.** To incentivize the *thinking-with-videos* behavior, we carefully curated the task distribution and data sources for the SFT dataset. It encompasses both single- and multi-turn trajectories, spans grounding and QA tasks, with diverse video durations. The detailed composition is shown in Fig. 3a. For data formatting, the reasoning path is enclosed in `<think>...</think>` followed by tool call parameters in `<tool_call>...</tool_call>`, while final answer outputs adopt the format `<answer>...</answer>`.

**RL Data.** To further enhance the model's capabilities in video grounding and videoQA, we employ a mixture of two data types during the RL stage. These data sources are designed to comprehensively cover the key dimensions of grounding and video reasoning. The detailed composition is shown in Fig. 3b. To ensure reliable reward signals for RL, all data annotations undergo rigorous human verification. For temporal grounding, low-quality samples are initially filtered, and the remainder undergoes careful human verification and correction. For long-video grounded QA data with both interval and answer annotations, annotators carefully inspect both, discarding ambiguous cases and correcting annotation errors to maintain high data fidelity. When a question can be answered from multiple temporal segments, annotators mark all valid evidence intervals whenever possible, so that alternative correct localizations are not unfairly penalized during RL.

**Data Balance.** To maintain generalization across diverse scenarios, we employ careful data balancing during training from three dimensions: (1) *Modality Balance*. We carefully regulate the proportion of video data while simultaneously maintaining some image data, which preserves general multimodal reasoning. (2) *Task Balance*. A diverse and balanced set of question types is included, *e.g.*, temporal grounding, temporal understanding, entity/action recognition, and relational reasoning. (3) *Video-length Balance*. We perform stratified sampling across a range of video durations. More details are presented in the Appendix A.

### 3.3. VideoTemp-Bench

To thoroughly examine the interaction between video grounding and understanding abilities, we conduct evaluations on video GQA datasets. However, existing video GQA benchmarks predominantly target short videos (<3min). To enable systematic evaluation of the model's GQA performance across varying temporal scales, we introduce VideoTemp-Bench. This benchmark segments videos into four duration-based categories: 0~3 minutes, 3~10 minutes, 10~20 minutes, and >20 minutes, with 300 samples in each category and 1,200 samples in total. It is designed to reveal how video understanding performance, *i.e.*, both temporal localization and final answering, varies as video duration increases. Please refer to the Appendix A.

### 4. Method

**Overview.** We introduce our agentic *thinking-with-videos* model, namely `VideoTemp-o3`. We first present task formulation. Then we introduce training strategies including SFT in Sec. 4.1 and RL in Sec. 4.2.

**Task Formulation.** We treat *thinking-with-videos* as a *localize-crop-answer* process. For long videos where uniform sampling often fails to capture key evidence, agentic MLLMs can iteratively invoke tools, *i.e.*, video cropping and dense sampling, to retrieve relevant frames and then answer based on them. In this way, we can bypass the sparse evidence challenge in uniform sampling. Formally, given a video-question pair $(V, Q)$, the model first skims the video at a low sampling rate $s_0$. Then it engages in iterative interaction turns. At each turn, it generates a textual reasoning $T$, along with either a temporal interval $P$ or a final answer $A$. If $P$ is predicted, an external cropping module extracts the corresponding clip from $V$ at a higher rate $s_d > s_0$, *i.e.*, $C = \mathrm{Crop}(V, P, s_d)$. The cropped clip $C$ is appended to the context for the next turn. The interaction terminates either when the model outputs the answer $A$ or reaches the maximum number of turns $T_{max}$. As a result, each training sample $i$ can be represented as a multi-turn trajectory:

$$\tau_i = \{(V, Q); ([T_{i,1}, P_{i,1}, C_{i,1}], \cdots, [T_{i,t}, A_i])\}, \quad (1)$$

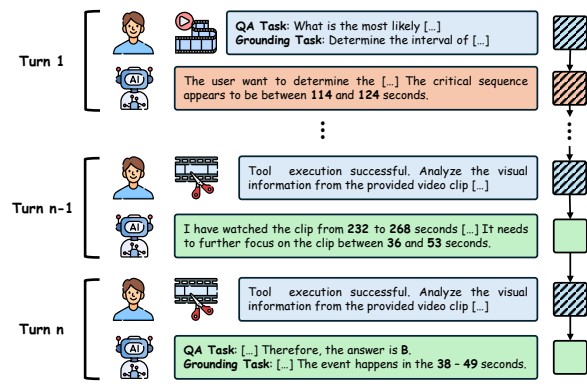

*Figure 4.* The unified masking mechanism, where only the last two turns of responses are supervised while others are masked.

where $t$ denotes the index of the final turn of this trajectory.

### 4.1. Cold Start Supervised Fine-Tuning

We begin by cold-start SFT to incentivize *thinking-with-videos* behaviors, where the model learns to first perform on-demand crop, then produce the final answer grounded in key visual evidence in the segments.

**Three Desirable Features.** To enable the model to solve complex problems flexibly and effectively, we summarize the following properties: (1) *On-demand cropping.* For short videos, the model can efficiently answer directly without cropping, *i.e.*, $t = 0$ in Eq. (1). As a result, we include short VideoQA data without any cropping. (2) *Reflection mechanism.* For challenging long videos, the initial grounding $P_1$ may be inaccurate, the model could perform multiple refinement rounds, *i.e.*, $t \geq 3$. In this regard, we curate multi-tool-call training samples with grounding reflection trajectories. (3) *Unifying temporal grounding and VideoQA.* Our model supports both VideoQA and grounding, so the answer $A_i$ in Eq. (1) can be either an answer to the question or a temporal interval. We employ the same format in the multi-turn dialogue for two tasks. The unification enhances the model's intrinsic grounding ability, which is fundamental to *thinking-with-videos*. Accordingly, we adopt a mixture of VideoQA and grounding data for training. Please refer to Sec. 3.2 for more details.

**Unified Masking Strategy.** In the collected tool-call data, the penultimate turn (Turn n-1 in Fig. 4) contains the correct temporal interval with the key evidence, and the final turn outputs the final answer. The temporal intervals in earlier turns are typically imprecise for multi-tool-call data, supervising these intervals during SFT can introduce noise. Based on this insight, we apply the training loss only to the final two turns of the model outputs in the multi-turn dialogue, while all earlier generations and user inputs are masked so they do not affect gradients.

*Table 1.* Main comparison of long video understanding. Gemini-1.5-Pro and GPT-4o are included as closed-source model references. For fair comparisons, we ensure all methods use the same input resolution and maximum frames. [†] denotes reproduced results using official checkpoints. **Bold** and underline denotes the best and the second best results.

| Method | MLVU ≈ 651s M-Avg | VideoMMMU ≈ 506s | | | | VideoMME (w/o subtitle) ≈ 1018s | | | | LVBench ≈ 4101s Avg. |
|---|---|---|---|---|---|---|---|---|---|---|
| | | Adapt. | Compr. | Prec. | Avg. | Short | Medium | Long | Overall | |
| Gemini-1.5-Pro (Team et al., 2024) | - | 49.3 | 53.3 | 59.0 | 53.8 | 81.7 | 74.3 | 67.4 | 75.0 | 33.1 |
| GPT-4o (Hurst et al., 2024) | 54.9 | 55.6 | 62.0 | 66.0 | 61.2 | 80.0 | 70.3 | 65.3 | 71.9 | 30.8 |
| VILA-1.5-40B (Lin et al., 2024) | 44.2 | 32.6 | 30.6 | 38.6 | 34.0 | - | - | - | 60.1 | - |
| TimeMaker-8B (Chen et al., 2024b) | 49.2 | - | - | - | - | - | - | 46.4 | 57.3 | 41.3 |
| VideoXL-7B (Shu et al., 2025) | 45.5 | - | - | - | - | - | - | - | 55.5 | - |
| VideoRFT-7B (Wang et al., 2025a) | - | - | - | - | 51.1 | - | - | - | 59.8 | - |
| LongVA-7B (Zhang et al., 2024) | 41.4 | - | - | - | - | 61.1 | 50.4 | 46.2 | 52.6 | - |
| LongVT-7B-RFT (Yang et al., 2025) | - | 35.7 | 43.7 | 56.7 | 45.3 | - | - | - | - | 41.3 |
| Video-MTR-7B (Xie et al., 2025) | 48.4 | - | - | - | - | - | - | 51.0 | 59.0 | - |
| VideoChat-R1-7B (Li et al., 2025b) | 45.2[†] | 40.2[†] | 45.6[†] | 53.0[†] | 46.8[†] | 72.2 | 60.7[†] | 50.9[†] | 62.1 | 39.3[†] |
| Video-R1-7B (Feng et al., 2025) | 48.0[†] | **44.2**[†] | 46.0[†] | 67.3[†] | 52.4 | **74.1**[†] | 61.1[†] | 51.2[†] | 61.4 | 40.1 |
| Qwen2.5-VL-7B (Bai et al., 2025) | 45.2[†] | 35.9[†] | 36.1[†] | 57.6[†] | 43.2[†] | 69.8[†] | 59.2[†] | 50.8[†] | 59.9[†] | 39.2[†] |
| VideoTemp-o3-7B-SFT (Ours) | 49.5 | 39.3 | 46.4 | 60.4 | 48.7 | 72.0 | 59.2 | 50.2 | 60.6 | 39.6 |
| VideoTemp-o3-7B-RL (Ours) | **54.2** | 43.0 | **47.8** | **69.0** | **53.2** | 72.2 | **66.6** | **54.7** | **64.5** | **43.0** |

## 4.2. Agentic Reinforcement Learning

**Preliminary of GRPO.** We adopt the on-policy variant of the GRPO algorithm (Shao et al., 2024; Guo et al., 2025). For each training sample consisting of a video $V$ and a question $Q$, the current policy $\pi_\theta$ is used to sample $G$ multi-turn trajectories $\{\tau_1, \tau_2, \ldots, \tau_G\}$. The policy is optimized by maximizing the following objective:

$$\mathcal{J}_{GRPO}(\theta) = \mathbb{E}_{(V,Q) \sim D, \{\tau_i\}_{i=1}^G \sim \pi_\theta(\cdot|V,Q)}$$
$$\frac{1}{\sum_{i=1}^G |\tau_i|} \sum_{i=1}^G \sum_{j=1}^{|\tau_i|} \left( A_{i,j} - \beta \mathbb{D}_{KL}[\pi_\theta \| \pi_{ref}] \right), \quad (2)$$

where $|\tau_i|$ denotes the total number of tokens generated by the model in trajectory $i$, encompassing the thinking process, grounding outputs, and the final answer across all turns. After obtaining the final reward $\{r_i\}_{i=1}^G$ for each complete trajectory, the advantage term $A_{i,j}$ is normalized within the group and shared across all tokens $j$ within the same trajectory $\tau_i$: $A_{i,j} = \frac{r_i - mean(\{r_i\}_{i=1}^G)}{std(\{r_i\}_{i=1}^G)}$. The term $-\beta \mathbb{D}_{KL}[\pi_\theta \| \pi_{ref}]$ serves to constrain the deviation of the current policy $\pi_\theta$ from the reference policy $\pi_{ref}$. The reward $r_i$ measures the quality of the entire trajectory, including the effectiveness of grounding and reasoning content, as well as the correctness of the final answer.

**Reward Design.** To jointly optimize answer correctness, format adherence, and temporal grounding, we present a dedicated reward system consisting of three rewards.

**(1) Accuracy Reward.** It measures answer correctness. The reward is 1 only if the response matches the ground truth.

$$R_{accuracy} = \begin{cases} 1, & \text{answer} = \text{gt}, \\ 0, & \text{answer} \neq \text{gt}. \end{cases} \quad (3)$$

**(2) Format Reward.** This encourages the model to follow the required format throughout the multi-turn dialogue.

$$R_{format} = \begin{cases} 1, & \text{format match}, \\ 0, & \text{otherwise}. \end{cases} \quad (4)$$

**(3) Penalty-aware IoU Reward.** This is used to measure the quality of timestamp grounding. Let the model's predicted interval be $[t'_s, t'_e]$ and the ground truth interval be $[t_s, t_e]$, the grounding IoU is defined as:

$$R_{IoU} = \frac{|[t_s, t_e]| \cap |[t'_s, t'_e]|}{|[t_s, t_e]| \cup |[t'_s, t'_e]|}. \quad (5)$$

When using IoU alone as a reward, the model may localize arbitrarily to hack the reward. To address this, we explicitly penalize these low-quality grounding behaviors. Specifically, when IoU is lower than a threshold $\sigma$, a penalty term $\lambda$ is applied to the response:

$$R_{penalty\text{-}IoU} = \begin{cases} R_{IoU} - \lambda, & R_{IoU} < \sigma, \\ R_{IoU}, & R_{IoU} \geq \sigma. \end{cases} \quad (6)$$

This design effectively mitigates reward hacking and encourages the model to produce reliable groundings while discouraging arbitrary guesses, enabling more stable learning of multi-turn grounding and refinements.

## 5. Experiment

### 5.1. Experimental Setup

**Implementation Details.** Following prior works, we choose Qwen2.5-VL-7B (Bai et al., 2025) as the backbone. The training pipeline is built upon *ms-swift* (Zhao et al., 2024)

*Table 2.* Main comparison of temporal grounding. **Bold** and underline denotes the best and the second best results.

| Method | Charades-STA | | ActivityNet-MR | |
|---|---|---|---|---|
| | R@0.7 | mIoU | R@0.7 | mIoU |
| VTimeLLM-7B (Huang et al., 2024) | 11.4 | 31.2 | 14.3 | 30.4 |
| Momentor-7B (Qian et al., 2024) | 11.6 | 28.5 | 12.4 | 29.3 |
| ChatVTG-7B (Qu et al., 2024) | 15.8 | 34.8 | 9.4 | 27.2 |
| TimeMaker-8B (Chen et al., 2024b) | 26.9 | 48.4 | **33.0** | **49.5** |
| ZoomV-7B (Pan et al., 2025) | 24.5 | 48.6 | 26.1 | 43.9 |
| VideoChat-R1-7B (Li et al., 2025b)† | **37.9** | 54.9 | 18.2 | 36.8 |
| Video-R1-7B (Feng et al., 2025)† | 18.1 | 35.1 | 6.2 | 18.3 |
| Qwen2.5-VL-7B (Bai et al., 2025)† | 18.8 | 38.5 | 10.4 | 26.8 |
| `VideoTemp-o3-7B-SFT` (Ours) | 28.0 | 48.9 | 24.7 | 41.1 |
| `VideoTemp-o3-7B-RL` (Ours) | 33.0 | **57.8** | 26.7 | 45.3 |

*Table 3.* Main comparison of video GQA. **Bold** and underline denotes the best and the second best results.

| Method | NextGQA | | ReXTime | |
|---|---|---|---|---|
| | mIoU | Acc | mIoU | Acc |
| VTimeLLM-7B (Huang et al., 2024) | - | - | 20.1 | 36.1 |
| TimeChat-7B (Ren et al., 2024) | - | - | 11.6 | 40.0 |
| DeepVideo-R1-7B (Park et al., 2025) | **36.8** | 72.5 | - | - |
| VideoChat-R1-7B (Li et al., 2025b)† | 32.4 | 70.6 | 10.3 | 71.1 |
| Video-R1-7B (Feng et al., 2025)† | 17.5 | 74.3 | 7.3 | 70.5 |
| Qwen2.5-VL-7B (Bai et al., 2025)† | 22.7 | 74.8 | 8.7 | 73.4 |
| `VideoTemp-o3-7B-SFT` (Ours) | 30.3 | 75.4 | 23.7 | 71.4 |
| `VideoTemp-o3-7B-RL` (Ours) | 33.4 | **76.4** | **29.5** | **74.4** |

and *vLLM* (Kwon et al., 2023), with customized support for multi-round tool-call interactions. In Eq. 6, we set the hyper-parameters $\lambda = 0.1$ and $\sigma = 0.1$. For additional training details, please refer to Appendix B.1.

**Benchmarks.** We adopt a diverse set of public benchmarks for each task. For long video understanding, we evaluate on MLVU (test set) (Zhou et al., 2025), VideoMMMU (Hu et al., 2025), VideoMME (Fu et al., 2025) and LVBench (Wang et al., 2025c), which assess multimodal reasoning and comprehension over long video. For temporal grounding, we use Charades-STA (Gao et al., 2017) and ActivityNet-MR (Krishna et al., 2017), which test the model's ability to localize temporal segments corresponding to text captions. For video GQA, we leverage NextGQA (Xiao et al., 2024) and ReXTime (Chen et al., 2024a) to evaluate the ability to answer fine-grained, temporally grounded questions. Besides, we introduce VideoTemp-Bench to analyze performance across varying video lengths.

## 5.2. Main Comparative Results

**`VideoTemp-o3` achieves remarkable long video understanding.** As shown in Tab. 1, `VideoTemp-o3` achieves state-of-the-art performance across nearly all long video understanding benchmarks. For example, our method obtains 2.4% and 1.7% improvements on VideoMME and LVBench, respectively, indicating strong multimodal understanding in long videos. The initial SFT stage incentivizes fundamental patterns for basic video understanding using curated trajectories, and subsequent RL consistently enhances video reasoning, *i.e.*, by roughly 4∼5% across all benchmarks.

**`VideoTemp-o3` exhibits strong internal temporal grounding abilities.** `VideoTemp-o3` also delivers strong temporal grounding performance, even comparable to that of expert grounding models like TimeMaker (Chen et al., 2024b), as shown in Tab. 2. The reliable grounding is a prerequisite for the implementation of *thinking-with-videos*.

*Table 4.* Ablations on training components in the SFT stage and rewards in the RL stage in comparison with our full model.

| ID | Variants | VideoMMMU | VideoMME | LVBench | ReXTime | |
|---|---|---|---|---|---|---|
| | | Overall | Overall | Overall | mIoU | Acc |
| (a) | `VideoTemp-o3` | **53.2** | **64.5** | **43.0** | 29.5 | 74.4 |
| (b) | w/o grounding data | 52.5 | 63.0 | 42.0 | 13.0 | 73.3 |
| (c) | w/o unified masking | 47.9 | 61.5 | 41.2 | 18.8 | 70.6 |
| (d) | w/o IoU reward | 51.6 | 63.3 | 41.7 | 26.2 | 73.7 |
| (e) | w/o penalty-aware | 44.2 | 63.7 | 40.7 | 23.8 | 73.6 |

**The *thinking-with-videos* paradigm ensures strong GQA performance.** As indicated in Tab. 3, `VideoTemp-o3` attains top-tier performance on both NextGQA and ReXTime. Notably, the model consistently attains higher mIoU and answer accuracy, indicating more precise evidence grounding and highlighting how improved localization contributes to stronger video understanding. The consistent gains validate the effectiveness of the on-demand tool-call abilities in *thinking-with-videos*. By identifying and focusing on the most relevant video segments, the model significantly enhances long video perception and reasoning.

## 5.3. Ablation Studies

**Ablations on grounding training data.** As Tab. 4 shows, SFT training with grounding data not only directly enhances grounding performance but also implicitly improves video QA accuracy. For example, (a) outperforms (b) 16.5% and 1.1% on mIoU and Acc of ReXTime, respectively.

**Ablations on unified masking mechanism.** Removing the unified masking strategy and instead supervising all responses during SFT leads to significant performance degradation, *i.e.*, (a) → (c) in Tab. 4. We attribute this degradation to the noise introduced by unmasked, incorrect reasoning paths of the initial coarse grounding turn. These results underscore the importance of our unified masking strategy in stabilizing training using correct signals.

*Table 5.* Performance of temporal grounding (mIoU) and VideoQA (Acc) of various durations in VideoTemp-Bench.

| Method | 0~3min | | 3~10min | | 10~20min | | > 20min | | Overall | |
|---|---|---|---|---|---|---|---|---|---|---|
| | mIoU | Acc | mIoU | Acc | mIoU | Acc | mIoU | Acc | mIoU | Acc |
| Gemini-2.5-Pro (Comanici et al., 2025) | 39.1 | 81.6 | 46.1 | 91.3 | 36.1 | 90.0 | 14.8 | 76.1 | 34.0 | 84.7 |
| VideoChat-R1-7B (Li et al., 2025b) | 25.2 | **82.0** | 6.7 | 87.6 | 4.7 | 73.6 | 1.8 | 53.6 | 9.6 | 74.2 |
| Video-R1-7B (Feng et al., 2025) | 13.3 | 81.0 | 4.0 | 86.0 | 4.0 | 75.3 | 2.1 | 53.3 | 5.9 | 73.9 |
| Qwen2.5-VL-7B (Bai et al., 2025) | 17.6 | 75.0 | 9.1 | 81.6 | 6.9 | 70.3 | 1.0 | 49.1 | 8.7 | 69.1 |
| VideoTemp-o3-7B-SFT (Ours) | 33.7 | 78.3 | 18.6 | 79.5 | 14.6 | 68.5 | 3.3 | **54.6** | 17.8 | 70.5 |
| VideoTemp-o3-7B-RL (Ours) | **35.3** | 81.0 | **32.0** | 90.6 | **21.7** | 81.3 | **4.0** | **54.6** | **23.4** | **77.0** |

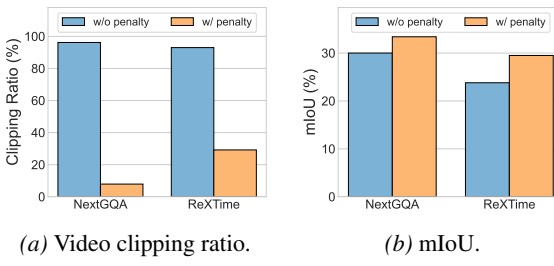

*(a)* Video clipping ratio.  *(b)* mIoU.

*Figure 5.* Reward hacking with native IoU rewards.

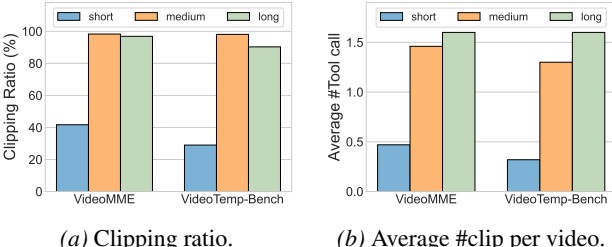

*(a)* Clipping ratio.  *(b)* Average #clip per video.

*Figure 6.* On-demand tool-call according to video length.

**Naive IoU reward can readily induce reward hacking.** Replacing the penalty-aware IoU reward $R_{\text{penalty-IoU}}$ with a standard IoU reward $R_{\text{IoU}}$ in Eq. (5) leads to performance degradation in Tab. 4. We delve into this phenomenon in Fig. 5. $R_{\text{IoU}}$ causes the tool-call ratio to rise sharply (Fig. 5a) while substantially degrading grounding quality (Fig. 5b). In short, $R_{\text{IoU}}$ without any penalty encourages the model to localize blindly to hack IoU rewards while neglecting grounding performance.

**Ablations on IoU reward.** In the RL stage, removing all IoU rewards (only keeping $R_{\text{acc}}$ and $R_{\text{format}}$) in Tab. 4 (d) yields a modest decline in both tasks, *e.g.*, a 1.6% decline in VideoMMMU. The effect of the IoU reward is similar to the grounding data in SFT, *i.e.*, enhancing the model's internal grounding abilities for desirable *thinking-with-video* behaviors.

### 5.4. Further Analysis

**Our method could perform on-demand tool-call.** We explicitly measure the tool-call statistics of VideoTemp-o3 across various video durations. Fig. 6a shows the tool-call rate for medium and long videos is significantly higher than that for short videos. This trend is accompanied by a corresponding increase in the average number of cropping operations (Fig. 6b). These results collectively indicate that, for longer videos, where informative cues are sparser, our model is more inclined to invoke video clipping to localize relevant visual evidence. In this regard, our method is able to perform on-demand clipping flexibly.

**Temporal grounding remains a key bottleneck for long**

**video understanding.** In Tab. 5, we evaluate several models on our VideoTemp-Bench containing videos of various durations. Generally, on longer videos (*e.g.*, >20min) where visual evidence is sparser, grounding performance drops remarkably, and QA accuracy deteriorates accordingly. For example, when video duration increases from <10min to >20min, the performance Qwen2.5-VL drops from 70.3% to 49.1%. We hypothesize that this degradation mainly comes from the sparse initial sampling required by context-length and computational constraints: if the key evidence is not observed during the initial skim, the model may fail to identify a reliable region for later refinement. In addition, hour-long videos often contain redundant scenes and transitions that further distract temporal localization. Our *thinking-with-videos* paradigm alleviates this issue to some extent. Nevertheless, accurate temporal localization remains a critical challenge for long videos. To further investigate this issue, we conducted additional experiments analyzing the effect of using ground truth video segments. Details can be found in Appendix C.

**Beyond grounding, certain types of tasks remain a major weakness for video understanding.** We comprehensively evaluate video understanding across a wide range of tasks in Fig. 7. The results show substantial performance variability across different video tasks. Generally, complex reasoning tasks (temporal reasoning) and fine-grained perception (counting and OCR) are more challenging, whereas basic perception tasks (information synopsis and recognition) are comparatively easier. Specifically, there is an approximately 40% gap between counting and recognition tasks.

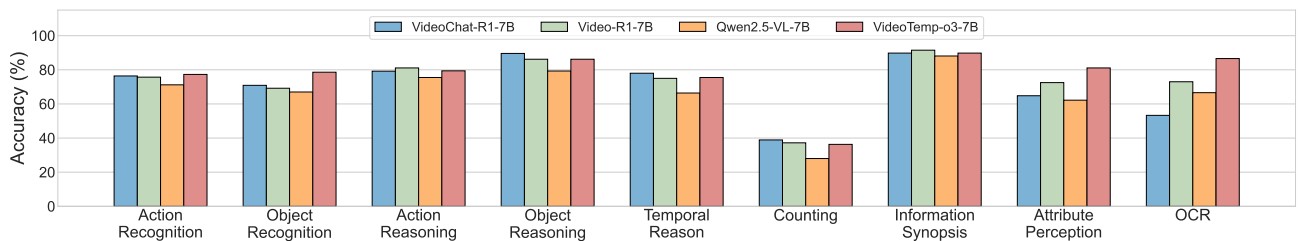

*Figure 7.* Performance of different video tasks in VideoTemp-Bench.

*Table 6.* Inference-time control comparisons on VideoMME (w/o subtitle). The first two controls are applied to Qwen2.5-VL-7B, while the next two controls are applied to `VideoTemp-o3`.

| Method | Short | Medium | Long | Overall |
|---|---|---|---|---|
| Prompt-only re-grounding | 69.0 | 56.3 | 49.8 | 58.4 |
| Inference-time self-correction | 64.7 | 63.4 | 53.2 | 60.5 |
| Qwen2.5-VL-7B | 69.8 | 59.2 | 50.8 | 59.9 |
| w/ crop once | 72.0 | 64.0 | 53.1 | 63.0 |
| w/o tool call | 70.3 | 63.4 | 52.7 | 62.1 |
| `VideoTemp-o3` | **72.2** | **66.6** | **54.7** | **64.5** |

**Learned tool-use behavior is more effective than fixed or prompt-only strategies.** To further isolate the effect of our agentic inference process, we compare `VideoTemp-o3` with several inference-time controls in Tab. 6. The first two control strategies are applied to Qwen2.5-VL-7B, which does not natively include a localize-clip-answer process. Prompt-only re-grounding and inference-time self-correction show only marginal differences from direct answering, indicating that prompt-level interventions alone provide limited gains. The latter two controls are applied to `VideoTemp-o3`: always cropping once and directly answering both underperform the full model, especially on medium and long videos where relevant evidence is more sparse. These results indicate that explicit trajectory training and reward optimization are important for learning when to crop, when to refine, and when to answer.

## 6. Conclusion

In this work, we study the agentic *thinking-with-videos* paradigm for long video understanding and propose a unified framework, namely `VideoTemp-o3`. It follows the pipeline of *localize-clip-answer*, by proactively cropping query-relevant video segments, the model could produce reliable answers grounded in key visual features. `VideoTemp-o3` is characterized by three features: on-demand video cropping, a reflection mechanism to refine imprecise grounding, and the unification of both temporal grounding and VideoQA tasks. To achieve this, we propose a unified masking mechanism in the SFT stage to incentivize desirable behaviors and devise dedicated rewards to

mitigate reward hacking. Besides, we introduce a data curation pipeline to curate high-quality multi-turn trajectories for SFT and reliable long video GQA annotation for RL. Owing to these components, our method outperforms previous methods across a broad spectrum of long video QA and grounding benchmarks. Future work could incorporate a broader suite of external tools, such as search engines and spatiotemporal grounding operations, to enable the model to tackle complex and up-to-date tasks in pragmatic scenarios.

## 7. Limitations

Although `VideoTemp-o3` improves long-video understanding through iterative temporal grounding, several limitations remain. For very long videos, the initial skim is still sparse due to context-length and computational constraints, so key evidence may be missed before refinement. This issue is more severe for extremely short events that last only a few frames. In addition, the localize-clip-answer process introduces sequential inference overhead, and the data construction pipeline still benefits from strong teacher models and verified temporal annotations, which may increase construction cost. Our unified masking strategy also assumes that later verified turns are more reliable than earlier exploratory turns; if future data construction can provide accurate supervision for every turn, this heuristic may need to be adapted. Future work could explore hierarchical sampling, finer temporal modeling, parallel clip verification, and weaker forms of grounding supervision.

## Acknowledgements

This work was supported in part by the National Natural Science Foundation of China (Nos. 62376140, 62376137, U23A20315, and 62572282), the Shandong Provincial Natural Science Foundation (No. ZR2022YQ59), the Science and Technology Innovation Program for Distinguished Young Scholars of Shandong Province Higher Education Institutions (No. 2023KJ128), the Special Fund for Taishan Scholar Project of Shandong Province, the Key R&D Program of Shandong Province, China (No. 2025CXGC020101), and Kuaishou (No. DJHL-20240801-004), providing essential funding and valuable institutional support to this research.

## Impact Statement

This paper aims to advance long-video understanding by improving temporal grounding and evidence-based video question answering. Such techniques may support beneficial applications such as video retrieval, educational content analysis, accessibility tools, and assistance for reviewing long-form visual records. At the same time, more capable video understanding systems may also be misused for intrusive surveillance, privacy-sensitive profiling, or automated decision making without appropriate human oversight.

Our method can still produce incorrect temporal localizations or answers, especially for very long videos, ambiguous questions, or short-duration events. Therefore, it should not be directly deployed in high-stakes scenarios where incorrect video interpretation may cause harm. In addition, the data construction process relies on strong teacher models and human verification, and may inherit biases from source datasets or teacher annotations. We encourage future work to study robustness, privacy protection, bias mitigation, and human-in-the-loop verification when applying long-video understanding models in real-world settings.

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

# A. Further Details about Dataset and Benchmark

To provide a more comprehensive understanding of the training data and VideoTemp-Bench, we present the duration distributions of various data subsets in Fig. 8.

As shown in Fig. 8a and Fig. 8b, the durations of SFT data without tool calls are concentrated in shorter video ranges. This aligns with the intuition that short videos often contain sufficient visual context, making explicit localization unnecessary. In contrast, longer videos require the model to process and filter more complex visual content, which motivates the construction of multi-turn tool-call samples specifically for such cases. For the RL phase, as shown in Fig. 8c, we adopt a diverse distribution of video durations. This broader coverage ensures the model is exposed to various temporal complexities and better equipped to learn on-demand tool invocation strategies in different video comprehension scenarios.

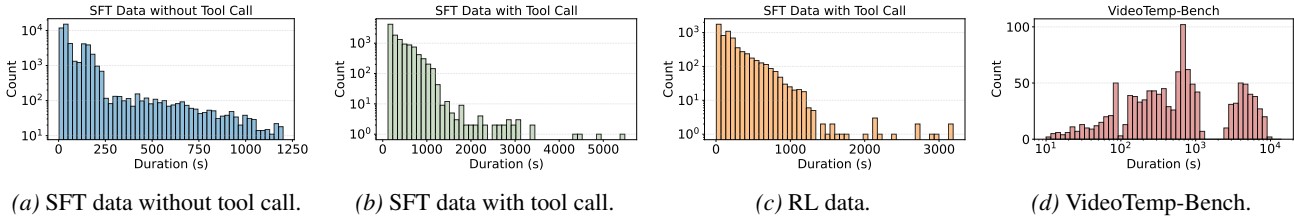

(a) SFT data without tool call.   (b) SFT data with tool call.   (c) RL data.   (d) VideoTemp-Bench.

Figure 8. Duration distribution of training data and benchmark.

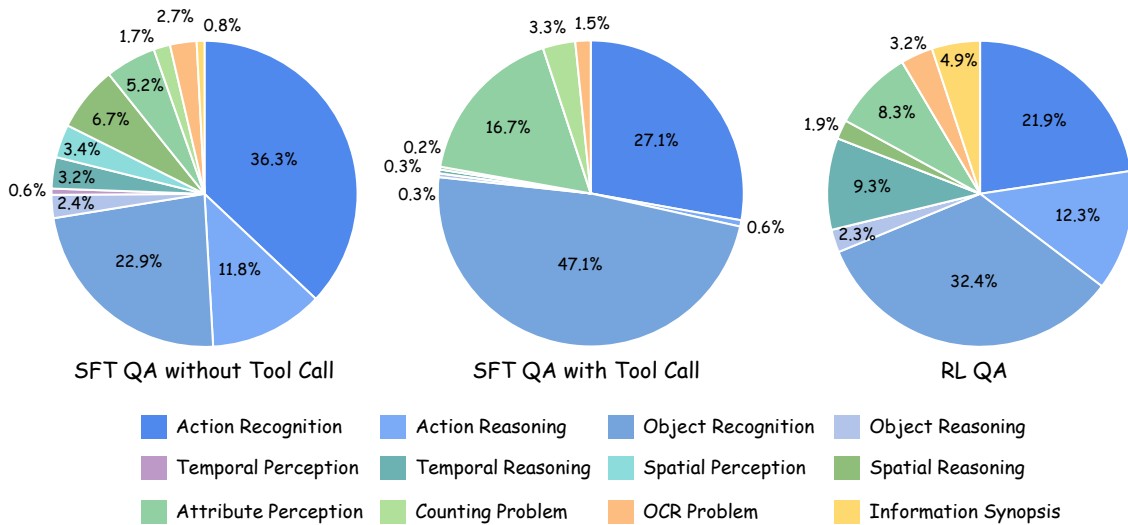

Figure 9. Distribution of question type in QA training data.

Beyond video duration distribution, we also present the question type distribution across QA datasets, as illustrated in Fig. 9. For the SFT QA data without tool calls, we incorporate a diverse range of question types to enhance the model's general comprehension and reasoning capabilities. In contrast, the SFT QA data with tool calls primarily focuses on question types such as Action/Object Recognition and Attribution Perception, which heavily rely on grounding within specific video segments. This targeted design aims to better train and strengthen the model's ability to perform *thinking-with-videos* reasoning. In the RL data, while maintaining a high proportion of the aforementioned grounding-focused categories, we further introduce a broader variety of question types. This diversity encourages the model to autonomously explore, adopt, and apply the *thinking-with-videos* paradigm more effectively throughout the RL process.

Regarding VideoTemp-Bench, as mentioned in Sec. 3.3, the benchmark is composed of four video length categories: 0∼3 minutes, 3∼10 minutes, 10∼20 minutes, and over 20 minutes. The duration distribution for this benchmark is detailed in Fig. 8d. Specifically, samples in the 0∼3 minute range are drawn from NextGQA and LongVILA; the 3∼10 minute and 10∼20 minute intervals utilize data from LongVILA and LongVideo-Reason (Chen et al., 2025b); and videos longer than 20 minutes are sourced from ScaleLong (Ma et al., 2025a). Each duration group contains 300 samples, totaling 1,200 samples.

*(a)* Training configurations.

| Configuration | SFT | RL |
|---|---|---|
| Learning rate | 1e-5 | 5e-6 |
| Batch size | 256 | 72 |
| Epochs | 3 | 1 |
| Group size | - | 8 |
| Max turns | 3 | 3 |
| Max frames | 512 | 512 |
| Max pixels | 224×224 | 224×224 |
| Tool call enabled | Yes | Yes |
| Training mode | full | full |

*(b)* Evaluation configurations for each benchmark.

| Benchmark | Max Frames | Max Pixels |
|---|---|---|
| MLVU (Zhou et al., 2025) | 256 | 448×448 |
| VideoMMMU (Hu et al., 2025) | 256 | 448×448 |
| VideoMME (Fu et al., 2025) | 1024 | 224×224 |
| LVBench (Wang et al., 2025c) | 1024 | 224×224 |
| Charades-STA (Gao et al., 2017) | 256 | 448×448 |
| ActivityNet-MR (Krishna et al., 2017) | 1024 | 224×224 |
| NextGQA (Xiao et al., 2024) | 256 | 448×448 |
| ReXTime (Chen et al., 2024a) | 1024 | 224×224 |
| VideoTemp-Bench (Ours) | 1024 | 224×224 |

*Table 7.* Configurations used in training and evaluation.

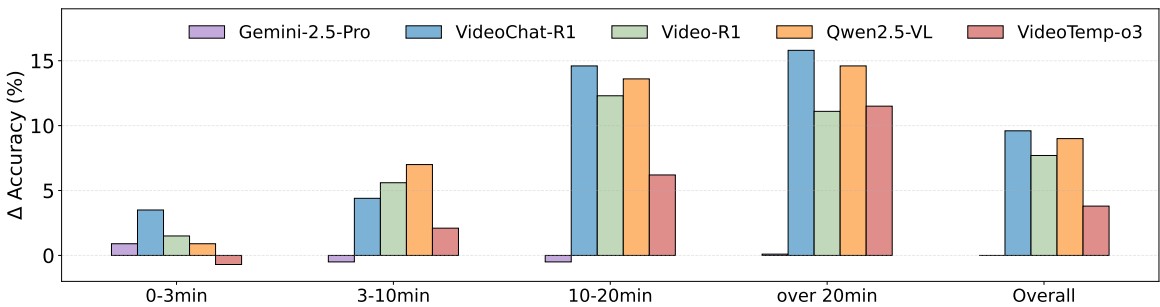

*Figure 10.* Accuracy change across video lengths when using ground truth video segments.

This stratified sampling design ensures both broad coverage and comparability, enabling stable and interpretable evaluation of model performance across varying video lengths. Compared with CG-Bench (Chen et al., 2025a), which focuses on clue-grounded QA for long videos, VideoTemp-Bench explicitly evaluates both temporal localization and answer accuracy across a wider range of video durations.

## B. Experimental Details

### B.1. Training Details

For SFT and RL, we set the learning rates to 1e-5 and 5e-6, respectively. All videos are sampled at a maximum rate of 2 FPS, with each original video uniformly sampled to up to 512 frames, and each clipped video sampled to a maximum of 64 frames. Each frame is constrained to a maximum resolution of 224×224 pixels. We use a batch size of 256 for SFT. For RL, each training step involves 72 samples, with each sample consisting of a group of 8 trajectories using GRPO. All the training is conducted on eight GPUs. For more details, please refer to Tab. 7a.

### B.2. Evaluation Details

During benchmark evaluation, videos are sampled at a maximum of 2 FPS, and the generation temperature is set to 0.1. Each dialogue session supports up to 3 conversational turns, allowing for one round of timestamp refinement. The detailed evaluation configurations for each benchmark are summarized in Tab. 7b.

## C. Impact of temporal grounding on VideoQA

To better understand the role of temporal grounding in VideoQA, we conduct an analysis by replacing the original videos in VideoTemp-Bench with the ground truth video segments corresponding to the annotated timestamps. This allows us to isolate and evaluate the effect of precise temporal grounding on model performance.

As shown in Fig. 10, Qwen2.5-VL, VideoChat-R1, and Video-R1 exhibits a consistent performance improvement across all video length categories when evaluated using the ground truth video segments. The performance gains become more

pronounced with increased video length, suggesting that longer videos pose greater challenges for segment localization. Consequently, access to accurately cropped content plays a more critical role in enabling precise reasoning and answer generation in such cases.

In comparison, `VideoTemp-o3` also achieves performance improvements under ground truth localization. However, the magnitude of this improvement is much smaller. Particularly for short videos (0-3 minutes), the gains are minimal. This can be attributed to `VideoTemp-o3`'s internalization of the *thinking-with-videos* paradigm, which enables it to effectively identify and reason over relevant temporal segments during inference. As a result, its dependence on externally provided localization is reduced, especially in shorter videos where the temporal space is inherently limited.

In contrast, Gemini-2.5-Pro shows negligible performance difference across all video length. This is likely due to its large context window and strong video understanding capabilities. Even with hour-long video inputs, Gemini-2.5-Pro can attend to nearly the entire sequence and accurately localize key information without the need for explicit cropping. This highlights the importance of dense frame sampling in long-form video understanding, which allows powerful models to maintain high reasoning accuracy across an extended temporal context.

## D. Reward Dynamics During RL Training.

To more precisely evaluate the significance of the penalty-aware IoU reward and the role of grounding data, we visualize the evolution of total training rewards and IoU-based rewards throughout the RL phase. These reward dynamics serve as direct indicators of the model's progress in acquiring the *thinking-with-videos* capability.

As illustrated in Fig. 11a and Fig. 11b, when using the penalty-aware reward, both the total training reward and IoU reward exhibit a consistently stable upward trend. This indicates that model is effectively learning temporal grounding and multi-step reasoning capabilities over the course of RL. In contrast, using a standard IoU reward (i.e., without penalty) results in a relatively flat IoU reward curve and even a decline in the total training reward. This suggests that the model struggles to reliably localize relevant video segments under this setting, leading to diminished learning effectiveness. Moreover, we observe a higher rate of format collapses, which not only degrade reasoning accuracy but also negatively affect performance on downstream tasks. In the third configuration, where grounding data is entirely omitted, the training reward initially appear relatively high following SFT. This is likely due to the model's prior learning from pure QA data, which focuses more on general reasoning. However, both the total training reward and the IoU reward deteriorate as training progresses. This decline is likely caused by the lack of grounding supervision, which impairs the model's ability to learn accurate localization behaviors during RL. Consequently, its overall *thinking-with-videos* capacity is undermined.

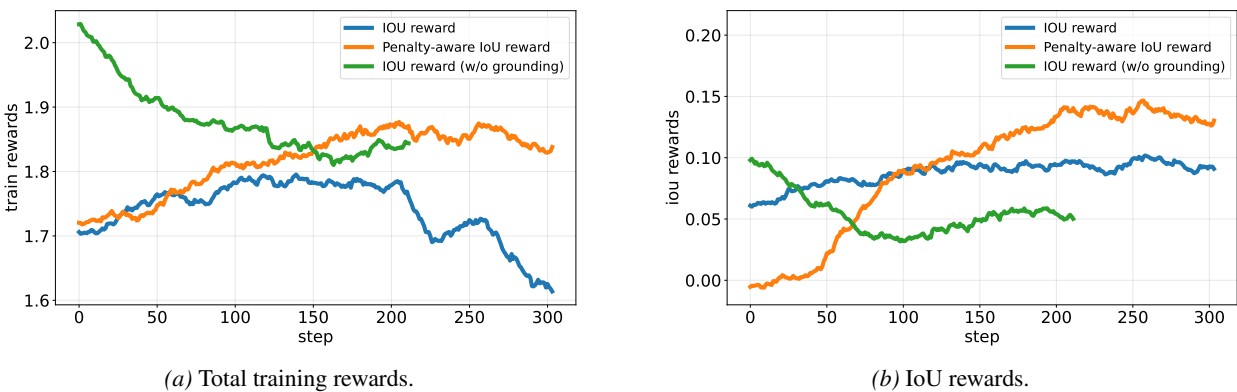

*(a)* Total training rewards.              *(b)* IoU rewards.

*Figure 11.* Trend of training rewards and IoU rewards.

## E. Open Source Declaration

This work uses the projects *ms-swift* and *vLLM*, released under the Apache License 2.0. We strictly adhere to the license requirements. The original projects' copyright notices and full license texts are available in their official repositories:

- `https://github.com/modelscope/ms-swift`
- `https://github.com/vllm-project/vllm`

## F. Prompt for Data Construction

This section presents the prompts used for constructing multi-turn data with tool calls. Specifically, Fig. 12 presents the prompt for temporal grounding, where the model is instructed to identify a relevant video segment given an input question. Fig. 13 shows the prompt used for question answering based solely on the cropped video segment. Fig. 14 illustrates the re-grounding prompt, which is employed when the grounding fails verification, prompting the model to refine its prediction using accumulated context. Finally, Fig. 15 depicts the prompt for final reasoning and answer generation, where the model is asked to provide a conclusive answer based on the full context, assuming the correct segment has been identified.

---

**Prior Input:**

[Orignal Video, Question]

--------------------------------------------------------------------------------

**Prompt for Temporal Grounding:**

You are an expert in video question localization. You will be given a video (duration: [duration] seconds) and a question that needs to be localized within it. Your task is to generate a progressive, step-by-step reasoning process in English to identify the precise timestamp range relevant to answering the question.

Your reasoning process must adhere to the following requirements:

1. Analyze and Reason: Begin by thoroughly analyzing the question and the video's content. Your primary goal is to identify a rough timestamp range where the answer to the question can be found. Before stating the final timestamp range, provide a detailed explanation of why this specific segment is relevant. Your reasoning should demonstrate a clear, progressive refinement of thought, connecting the video's visual and audio cues directly to the question.

2. Structured Approach: Your analysis should be logical and structured. Start by identifying the key events or visual elements needed to answer the question. Ground your observations firmly in the video's content.

3. Incorporate Self-Reflection: To make your thought process transparent, incorporate self-reflective phrases (e.g., "Let me think...", "Hmm, upon closer inspection...", "Wait, I should reconsider...") to validate and refine your conclusions as you narrow down the timestamp.

4. Timestamp Formatting: Crucially, all timestamps mentioned in your analysis must be in seconds. For example, convert "1 minute and 19 seconds" to "79 seconds".

5. Timestamp Span: The proposed timestamp range should be relevant and concise, generally between 5 and 60 seconds long.

6. Concluding Statement: Your final sentence must explicitly state the focused time range, including the start and end times.
- "The critical sequence appears to be between 52 and 65 seconds."
- "Therefore, the entire action is contained between roughly 40 and 60 seconds."
- "So, I should focus on the time in the restaurant, which is from 15 to 51 seconds."
- "I will focus my analysis on the first few seconds, from 0 to 10 seconds, to clearly identify its position."

7. JSON Output: The final output must be in JSON format. This format should clearly associate the identified timestamps with your corresponding detailed analysis. Ensure the start and end times of the final range are distinct, reflecting the refined reasoning process.

### Example Output ###
{
"think": "[think process]",
"timestamp": [[start_time, end_time]]
}

### Question ###
[question]

---

*Figure 12.* Prompt for temporal grounding.

**Prior Input:**
[Video Segment, Question]

**Prompt for QA on Clipped Video Segment:**

Please answer the question based on the provided video clip. Your output must be in JSON format.

- If the video clip provides enough information to answer the question, output the corresponding option (e.g., "A", "B", "C", "D").
- If the video clip does not provide enough information to answer the question, output `None`.

### Question ###
[question]

### Example Output 1 (Answerable) ###
{
  "answer": "[ABCD]"
}

### Example Output 2 (Unanswerable) ###
{
  "answer": None
}

*Figure 13.* Prompt for qa on clipped video segment.

---

**Prior Input:**
[Orignal Video, Question, Grounding Process, Video Segment]

**Prompt for Re-Grounding:**

The cropped video clip corresponding to the previously identified timestamp range is now provided. However, this clipped video segment failed to answer the question correctly. Please re-evaluate your prior analysis, conduct a new reasoning process, and provide a revised timestamp range in the same format.

Your new reasoning process must adhere to the following requirements:

1. Assume Self-Correction: Frame the re-evaluation as a self-discovered error, not as a correction prompted by the user. The analysis and reasoning should strictly follow the pattern demonstrated in the provided example.

2. Start your reasoning with "I have watched the clip from ..."

3. Analyze the Clipped Segment: First, describe the content of the previously provided video clip.

4. Reflect on the Failure: Explain why this specific segment was insufficient or incorrect for answering the original question.

5. Provide New Reasoning: Following the requirements from our previous interaction, conduct a new, step-by-step reasoning process to identify a more accurate timestamp range. This should include self-reflection and a clear, logical progression of thought that connects the video's content to the question.

Remember to present the final output in the specified JSON format, with all timestamps in seconds and a concluding sentence that clearly states the new time range.

### Example Output ###
{
  "think": "[think process]",
  "timestamp": [[start_time, end_time]]
}

### Question ###
[question]

*Figure 14.* Prompt for re-grounding.

**Prior Input:**

[Orignal Video, Question, Grounding Process, Video Segment, (Re-Grounding Process, Video Segment, …)]

----------------------------------------------------------------------------

**Prompt for Final Reasoning and Answer:**

The cropped video clip corresponding to the previously identified timestamp range is now provided. Your task is to conduct a detailed analysis of this clip and provide a final answer to the question. Integrate insights from your prior analysis with specific observations from this new clip.

Your reasoning process must adhere to the following requirements:

1. Focus on Visual Content: Base your analysis exclusively on the visual information within this specific video clip to ensure accuracy.

2. Timestamp Formatting: All timestamps in your analysis must be in seconds. For example, convert "1:19" to "79 seconds".

3. Step-by-Step Analysis:
- Systematically break down the events in the clip.
- Proceed with a logical, clear, and succinct explanation.
- Incorporate self-reflective reasoning (e.g., "Let me think...", "Upon closer review...", "Wait, I need to verify...") to demonstrate a refined thought process.

4. Output Format: The final output must be in JSON format, clearly linking specific timestamps (with distinct start and end times) to your corresponding analysis and culminating in the final answer.

### Example Output ###
{
"think": "[think process]",
"answer": "[ABCD]"
}

### Question ###
[question]

*Figure 15.* Prompt for final reasoning and answer.

## G. Case Study

To better visualize the behavior of `VideoTemp-o3`, we present its actual outputs in Fig. 16 to Fig. 21.

**Original Video (135s)**

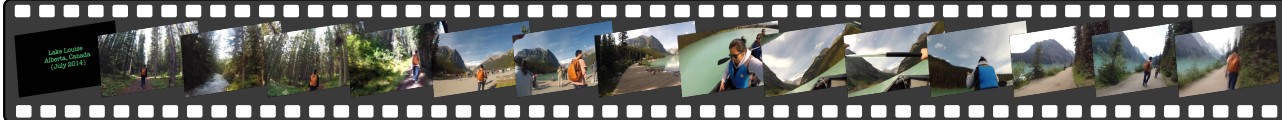

**Clipped Video (29s)**

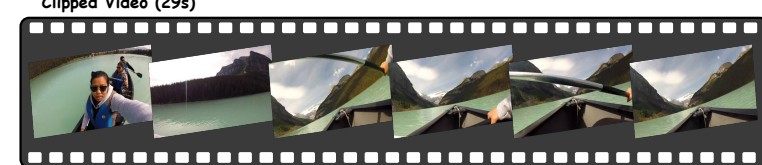

Determine the interval of "Three people ride a canoe near the base of the mountains."

Ground Truth: [67.57, 102.00]

**Turn 1: Grounding**

<think> I need to find the scene where three people are canoeing near mountains. I'll scan for this activity. The video shows hiking initially. Around 68 seconds, the camera perspective shifts to a selfie view inside a canoe with three people. They are paddling on a calm lake with mountains directly behind them. This action continues for several seconds. The sequence ends around 97 seconds when the view switches to a first-person perspective. Therefore, the event is contained in this segment. The critical sequence seems to be between 68 and 97 seconds. </think><tool_call>[68, 97]</tool_call>

**Turn 2: Answer**

<think> I am looking for the scene where three people are canoeing near mountains. The clip starts at 68 seconds with a selfie view of three people in a canoe on a lake. They are actively paddling, and large mountains are visible behind them. This action continues for the entire duration of the clip. The scene shifts to a first-person perspective at 97 seconds, but it is still on the water with mountains in the background. The entire sequence from 68 to 105 seconds shows the canoeing activity. The entire action seems to be contained between roughly 68 and 105 seconds..<think><answer>[68, 105]</answer>

*Figure 16.* Grounding task case of `VideoTemp-o3`.

**Original Video (394s)**

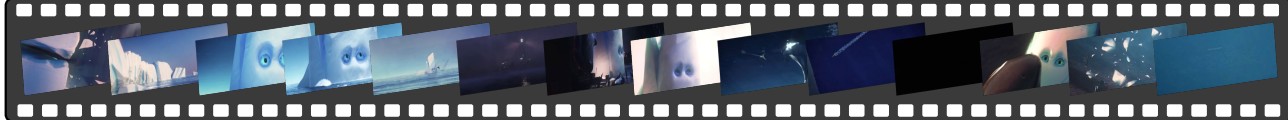

**Clipped Video (8s)**

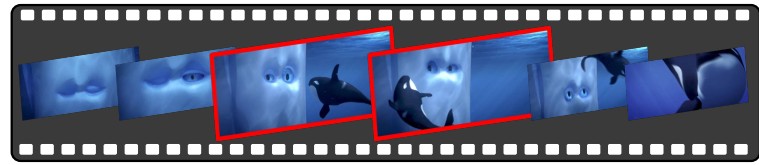

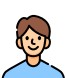

Who is the little iceberg meet first?

A. A little dolphin.          B. A little shark.
C. A little killer whale.     D. A little turtle.

**Turn 1: Grounding**

<think>The user wants to identify the first creature that the little iceberg meets in the video. I need to find the scene where the iceberg, which has eyes, comes into contact with another animal for the first time. I will scan the video for this interaction. The video starts with the iceberg's creation and growth. Around 48 seconds, the iceberg's eyes appear, giving it life. It then begins to melt. At approximately 1 minute and 20 seconds (80 seconds), the iceberg is shown underwater. Let me focus on this part. Hmm, upon closer inspection, at around 1 minute and 23 seconds (83 seconds), a black and white animal swims into the frame. This is clearly a killer whale, also known as an orca. The iceberg watches the whale curiously. This encounter is the first one depicted. The whale continues to swim around the iceberg for several seconds. Therefore, the first creature the iceberg meets is a killer whale. The relevant sequence starts when the whale appears and ends after they have interacted for a moment.</think><tool_call>[83, 90]</tool_call>

**Turn 2: Answer**

<think>I am analyzing the clip from 83 to 90 seconds to determine the first creature the iceberg meets. At the start of this segment, around 83 seconds, the camera shows a close-up of the iceberg's face underwater. A black and white fish, identifiable as a killer whale (orca), enters the frame from the left at 84 seconds. The iceberg's eyes follow the whale as it swims past. From 85 to 87 seconds, the whale circles the iceberg, and the iceberg watches it intently. There is no other creature present in this sequence. Based on this clear visual evidence, the first creature the iceberg meets is a killer whale. This confirms option C is correct.<think><answer>C</answer>

*Figure 17.* QA task case 1 of `VideoTemp-o3`.

**Original Video (1045s)**

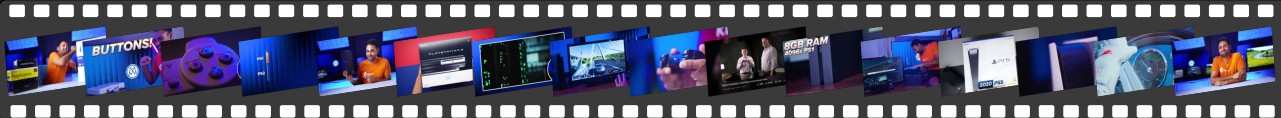

**Clipped Video (5s)**

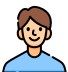

What is the brand of the TV used with PS2 in the video?

A. TOSHIBA.        B. BenQ.
C. SAMSUNG.        D. HITACHI.

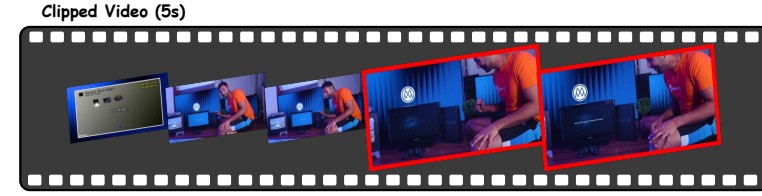

**Turn 1: Grounding**

<think>The user wants me to identify the brand of the TV used with the PlayStation 2 (PS2) in the video. I need to find the part of the video where the PS2 is unboxed or demonstrated on a TV, and then identify the TV's brand. 1. Scan for PS2 Segment: The video is about the evolution of PlayStation consoles. I'll skip ahead to the section where the PS2 is discussed. The PS2 section starts around 203 seconds. 2. Locate Unboxing/Display: At 203 seconds, the PS2 box is shown. Then, at 215 seconds, the presenter shows the contents of the box, which includes a memory card reader and cables. Following this, at 224 seconds, he takes out the console itself. 3. **Focus on TV Demonstration:** The next logical step would be showing the console in action. Ah, at 237 seconds, the video cuts to a shot of a television displaying a racing game. The game is shown running on two different TVs side-by-side.\n\n4. **Identify the Brand:** Let me look closely at the TV screens. On the left, there is a black television playing the game. This segment directly shows the brand of the TV used for the PS2 demonstration.</think><tool_call>[237, 241]</tool_call>

**Turn 2: Answer**

<think>I need to identify the brand of the TV used with the PS2 in the provided clip, which runs from 237 to 241 seconds. At 237 seconds, the clip begins with a split-screen view of a racing game. On the left side of the screen, a television is playing the game. Let me look closely at that TV. The brand name is clearly visible on the front bezel. It's a TOSHIBA. The model number is also partially visible. Since the question asks for the brand of the TV used, and the Toshiba TV is prominently featured on the left, it's the most likely candidate for the one being showcased. Therefore, the correct answer is A. TOSHIBA.<think><answer>A</answer>

*Figure 18.* QA task case 2 of `VideoTemp-o3`.

**Orginal Video (413s)**

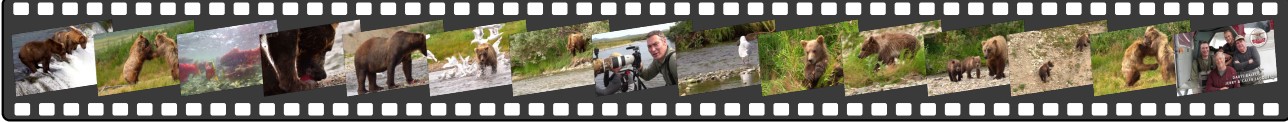

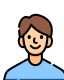

Why does the mother brown bear in the video make her cubs run to higher ground?

A. Because you can see further from the high ground.
B. Because she spotted two tiger that might kill the cubs.
C. Because she spotted two adult male brown bears that might kill the cubs.
D. Because it's easier to catch salmon from the high ground

**Clipped Video (6s)**

**Turn 1: Grounding**

<think>The user wants to know why the mother brown bear makes her cubs run to higher ground. I need to find the part of the video where this action occurs and understand the context. I will scan the video for scenes involving a mother bear, her cubs, and higher ground.\n\nFirst, I'll look for a mother bear with cubs. The video shows many such scenes. For example, at around 291 seconds, there is a clear shot of a mother bear and her two cubs on a rocky riverbank. \n\nNext, I need to find the specific moment she tells them to go to higher ground. I'll scrub through the video looking for a command or movement. Ah, at around 306 seconds, the mother bear stands up and begins to speak to her cubs. She gestures with her paws and then points towards the hillside behind them. At 307 seconds, she says something to them, and they immediately react. At 308 seconds, one cub runs ahead, and at 309 seconds, the other cub follows. The action starts at 306 seconds with her command and ends around 311 seconds after both cubs have run up the hill. This provides full context for the action.</think><tool_call>[306, 311]</tool_call>

**Turn 2: Answer**

<think>I need to analyze the provided clip (306-311 seconds) to determine why the mother brown bear tells her cubs to run to higher ground. \n\nAt 306 seconds, the clip shows a mother bear and her two cubs on a rocky riverbank. The mother stands up and gestures with her paw, clearly giving a command. \n\nAt 307 seconds, she speaks to her cubs, and they immediately react. \n\nAt 308 seconds, one cub runs ahead. \n\nAt 309 seconds, the other cub follows. \n\nThey all run up the hill behind them. \n\nThis sequence shows a clear and urgent movement away from the current location. In the context of bear behavior, especially with cubs, such a rapid and coordinated movement is typically a response to a perceived threat. The options must be evaluated based on this direct visual evidence. \n\nOption A (better visibility) is incorrect; the cubs are moving away from a vantage point, not to it. \n\nOption B (tigers) is incorrect as the setting is a bear habitat with no tigers present. \n\nOption D (easier salmon catching) is also incorrect; the cubs wouldn't be sent away while their mother is fishing. \n\nOption C (spotted two adult male brown bears) is the only plausible explanation. The urgency of the cubs' flight suggests they are fleeing from a predator. Since bears are the predators here, spotting other adult males would be a valid threat. The clip visually establishes the command and the subsequent flight, which is the standard reaction to a perceived predator threat.<think><answer>C</answer>

*Figure 19.* QA task case 3 of `VideoTemp-o3`.

**Orginal Video (313s)**

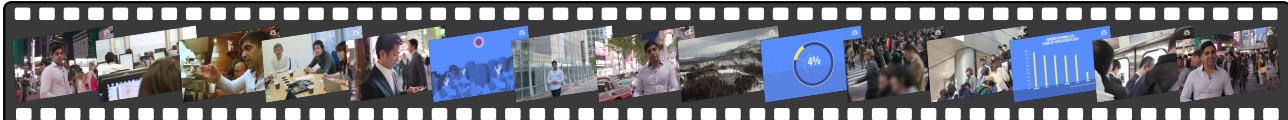

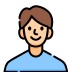
How many Japanese people would feel ashamed for taking paid leave according to the video?

A. 4%.     B. 36%.     C. 63%.     D. 42%.

**Clipped Video (7s)**

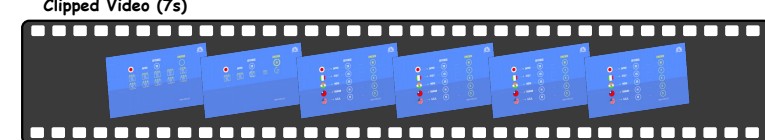

**Turn 1: Grounding**

\<think\>The user wants to know the percentage of Japanese people who feel ashamed for taking paid leave. I need to find the part of the video where this statistic is presented. I'll scan the video for infographics or on-screen text that discusses employee leave culture in Japan.\n\nFirst, I recall seeing some statistics about work culture early in the video. Around the 50-second mark, there's an animation showing a bar graph comparing countries' rates of unused vacation days. Let me check that section again. Ah, yes, at approximately 51 seconds, the video shows a pie chart with the text \"63% FEEL SHAMEFUL\". This seems directly relevant. Therefore, the critical information is contained within this short animation.\</think\>\<tool_call\>[51, 57]\</tool_call\>

**Turn 2: Grounding**

\<think\>I have watched the clip from 51 to 57 seconds. This segment shows an animated bar graph comparing the number of vacation days offered and unused across different countries. While it does show statistics related to work culture, it doesn't specifically focus on the percentage of Japanese people who feel ashamed about taking paid leave. My initial analysis was too broad and missed the more direct visual cue. Let me refine my search. Ah, right after the bar graph, another animation begins. At around 58 seconds, a pie chart appears with a shocked emoji and the text '63%'. This is followed by a transition to a man sweating at 101 seconds. The most direct and concise piece of evidence linking the feeling of shame to a specific percentage is the pie chart itself. Therefore, focusing on the moment the pie chart is introduced will be better. The critical sequence appears to be between 58 and 100 seconds.\</think\>\<tool_call\>[58, 100]\</tool_call\>

**Clipped Video (43s)**

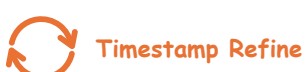
**Timestamp Refine**

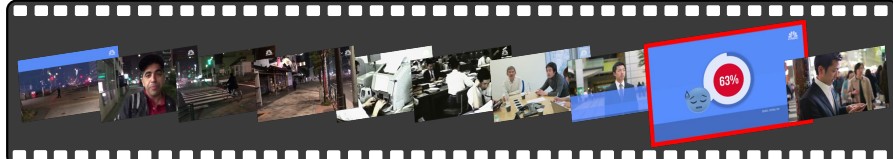

**Turn 3: Answer**

\<think\>I have watched the clip from 58 to 100 seconds. Initially, the clip shows various scenes of empty streets and office workers in Japan, illustrating the long hours they work. However, the most direct and explicit information regarding the percentage of people feeling ashamed is contained in the animated graphics that follow. Ah, at exactly 101 seconds, an animation begins with a shocked emoji appearing next to a pie chart. The text '63%' is prominently displayed over the pie chart. This graphic remains on screen until about 106 seconds. This is the precise moment where the statistic is shown. Therefore, the percentage of Japanese people who feel ashamed for taking paid leave is 63%, which corresponds to option C.\<think\>\<answer\>C\</answer\>

*Figure 20.* QA task case 4 of `VideoTemp-o3`.

**Orginal Video (502s)**

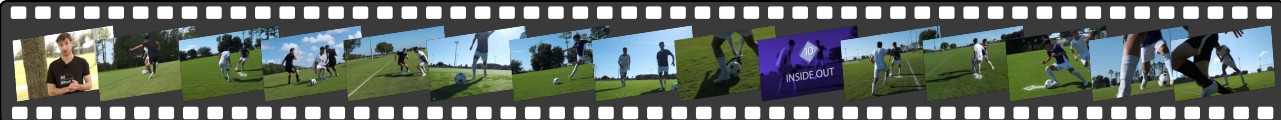

In which skill are there only three people on the field?

A. No.10 skill.    B. No.7 skill.
C. No.13 skill.    D. No.3 skill.

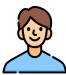

**Clipped Video (4s)**

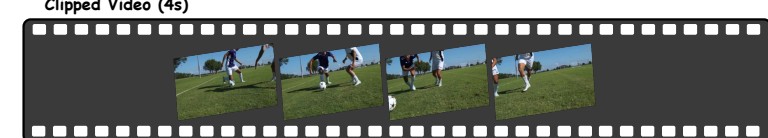

**Turn 1: Grounding**

<think>The user wants me to identify which soccer skill involves exactly three people on the field. I need to scan the video for title cards that introduce different skills and observe the number of players in the corresponding scenes.\n\nFirst, I'll skim through the video to find the title cards. The video is structured with numbered sections for each skill. I see title cards appearing periodically.\n\nLet's check each option:\n- Option A: \"No.10 skill.\" I need to find section 10.\n- Option B: \"No.7 skill.\" I need to find section 7.\n- Option C: \"No.13 skill.\" I need to find section 13.\n- Option D: \"No.3 skill.\" I need to find section 3.\n\nNow, let me focus on finding section 3. I recall seeing a title card for \"THE UNDERFOOT STOP\" which is likely section 3. Let me locate that part.\n\nHmm, upon closer inspection, the section titled \"THE UNDERFOOT STOP\" appears at approximately 142 seconds (2 minutes 22 seconds). In this specific segment, the video shows a wide shot of a soccer field where three players are clearly visible. One player in white is dribbling, while two other players in black are defending. This visual evidence directly supports the idea that this particular drill involves three people.\n\nTherefore, the correct answer should be related to section 3. The question asks about 'the' skill, implying one correct answer. Since the description points to section 3, the answer is linked to that specific section.</think><tool_call>[142, 145]</tool_call>

**Turn 2: Grounding**

<think>I have watched the clip from 142 to 145 seconds. This segment showed a wide shot of a soccer field where two players were present: one in white dribbling and one in black defending. My initial analysis mistakenly pinpointed two players but overlooked the specificity required to choose between the four distinct numbered options correctly. To rectify this, I must locate the precise introductory title card for one of these numbered drills, as the question asks about a specific numbered skill. To localize the search, I will evaluate the temporal proximity to the critical keyword phrase \"three people\". The introductory title card for section 3, \"THE UNDERFOOT STOP\", is visually identifiable at 91 seconds. Critically, just prior to this title card's appearance, at 90 seconds, the commentator says, \"...and the third one is the underfoot stop.\" This verbal cue establishes the context that the subsequent action will be linked to the 'third' numbered section. Therefore, the combination of the introductory title card and its contextual lead-in provides a much more accurate and sustained basis for identifying the correct numbered skill set. \n\nThe refined localization should encompass this contextual verbal cue followed by the explicit title card itself. Thus, the entire action is contained between roughly 90 and 94 seconds.</think><tool_call>[90, 94]</tool_call>

**Clipped Video (5s)**

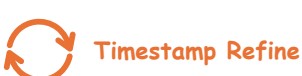

**Timestamp Refine**

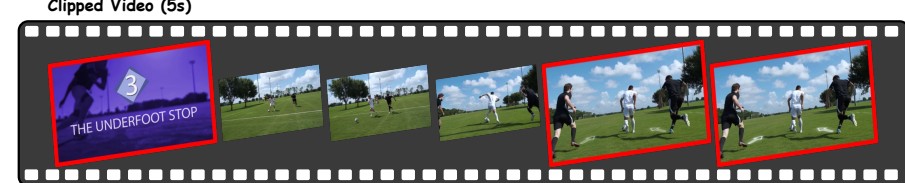

**Turn 3: Answer**

<think>I have watched the clip from 90 to 94 seconds. At 90 seconds, the commentator verbally introduces the third skill, saying, \"...and the third one is the underfoot stop...\". Immediately after, at 91 seconds, the video cuts to a title card that explicitly displays the number '3' inside a stylized diamond shape, followed by the text 'THE UNDERFOOT STOP'. This title card remains on screen until about 93 seconds. Following this, at 94 seconds, the video shows a wide shot of a soccer field where three players are visible: one in white dribbling, and two defenders in black. This confirms visually that this specific section is titled 'Number 3'. Comparing this to the given options, the content unequivocally points to the 'No.3 skill'. Therefore, option D is the correct description.<think><answer>D</answer>

*Figure 21.* QA task case 5 of `VideoTemp-o3`.

