# OpenReview forum: "VideoTemp-o3: Harmonizing Temporal Grounding and Video Understanding in Agentic Thinking-with-Videos"
_ICML.cc/2026/Conference — ICML 2026 regular_

### Official Review · Reviewer_nabb · 2026-02-22

**Soundness:** 3
**Presentation:** 3
**Significance:** 3
**Originality:** 3
**Overall Recommendation:** 4
**Confidence:** 2

**Summary:**

This paper propose VideoTemp-o3 for long video understanding, a unified framework that jointly models temporal grounding and video question answering (VideoQA) in a single architecture. Extensive experiments on standard long video understanding, temporal grounding, and video GQA benchmarks demonstrate state-of-the-art performance for the VideoTemp-o3, with ablation studies validating the efficacy of key components.

**Compliance With Llm Reviewing Policy:**

Affirmed.

**Final Justification:**

My concerns are largely addressed, and I decide remain my score. I encourage the authors to incorporate these additional results into the future version to further strengthen the paper.

**Key Questions For Authors:**

1. The penalty-aware IoU reward uses hyperparameters λ=0.1 and σ=0.1, but no sensitivity analysis is provided to justify these choices. How do varying values of λ (penalty strength) and σ (IoU threshold) impact the model’s performance? A detailed sensitivity analysis would strengthen the technical soundness of the reward design.
2. The data curation pipeline relies heavily on Gemini-2.5-Pro for re-grounding and filtering training samples, but the paper does not report the accuracy or consistency of Gemini-2.5-Pro’s annotations (e.g., IoU between Gemini-2.5-Pro’s predicted intervals and ground truth, error rate for multi-turn trajectories). See weakness.
3. VideoTemp-o3’s grounding performance drops sharply for ultra-long videos (>20min), but the paper does not explore much for that. Is this decline due to insufficient training data for ultra-long videos, context length limits of the Qwen2.5-VL-7B backbone, sparse visual evidence, or a failure of the iterative grounding pipeline to scale?
4. The model performs poorly on fine-grained tasks (counting, OCR) for long videos, but the framework only supports video cropping and dense sampling. Have you considered integrating external tools (e.g., OCR models, object detectors) into the agentic pipeline to address these gaps?

**Limitations:**

No, I think the author should discuss the model’s poor performance on videos >20min, because it is very critical for real senarios.

**Strengths And Weaknesses:**

Strengths: The paper demonstrates a well-validated methodology. The proposed VideoTemp-o3 is well grounded in "thinking-with-videos" paradigms, and each component is supported by empirical evidence. Key strengths include:
1. The authors use a consistent backbone (Qwen2.5-VL-7B) for fair comparisons with baselines, evaluate across diverse standard benchmarks (MLVU, Charades-STA, NextGQA), and introduce VideoTemp-Bench for long video evaluation.
2. Ablation studies systematically validate core components (grounding data, unified masking, IoU reward), and case studies illustrate the model’s iterative grounding performance.

Weaknesses:
1. Key hyperparameters for the penalty-aware IoU reward (λ=0.1, σ=0.1) are set without sensitivity analysis. The paper does not explore how varying these values affects grounding performance, leaving uncertainty about the optimality of the chosen settings.
2. The data pipeline relies heavily on Gemini-2.5-Pro for re-grounding and filtering, but the paper does not report the consistency or error rate of Gemini-2.5-Pro’s annotations. This introduces potential bias in training data quality, as the model’s performance may depend on the unvalidated accuracy of the external annotator.
3. The model’s poor performance on videos >20min is attributed to "sparse visual evidence" but not further investigated. The paper does not test hypotheses (e.g., insufficient training data for ultra-long videos, context length limits of the backbone), which can be guided for future improvements.

Other concerns:
1. Key information about VideoTemp-Bench (e.g., sample size per duration category, data sources) is only available in the appendix, not front-loaded in the main text.
2. The paper compares VideoTemp-o3 to general multimodal models (Gemini-1.5-Pro, GPT-4o) but does not explicitly frame these as "out-of-domain" baselines. This may confuse readers about whether these models are direct competitors.This is because they are not direct competitors, as they lack agentic "thinking-with-videos" capabilities.

---

> ### Author Rebuttal · Authors · 2026-03-30
>
> We sincerely appreciate the reviewer for spending valuable time in reviewing our paper and providing detailed comments. we hope that our response below will address your concerns.
>
> > W1 / Q1: Sensitivity of the penalty-aware IoU reward (λ=0.1, σ=0.1)?
>
> Thank you for the suggestion. Using VideoMME as an example, we previously conducted the following sensitivity analysis:
>
> |Setting|Short|Medium|Long|Overall|
> |-|-|-|-|-|
> |Qwen2.5-VL-7B|69.8|59.2|50.8|59.9|
> |w/o IoU|72.2|63.8|53.9|63.3|
> |λ=0,σ=0 (naive IoU)|72.9|64.5|53.7|63.7|
> |λ=0.1,σ=0|72.8|64.7|54.0|64.3|
> |λ=0.1,σ=0.1 (ours)|72.2|**66.6**|**54.7**|**64.5**|
> |λ=0.1,σ=0.2|**73.7**|62.8|52.9|63.2|
>
> As shown above, our chosen setting, λ=0.1, σ=0.1, gives the best overall trade-off, especially on medium and long videos. This supports the effectiveness of the selected hyperparameters. Moreover, other nonzero settings still outperform most baselines in the paper, suggesting that the overall benefit comes from introducing the penalty-aware IoU reward itself. We will include a more complete sensitivity table and discussion in the revised paper.
>
> > W2 / Q2: Gemini-2.5-Pro annotation accuracy and consistency.
>
> Benefiting from Gemini-2.5-Pro’s strong long-video understanding ability and very long context window, which allows it to process full long videos directly, the generated annotations are highly reliable; in our observation, they are even better than annotations in many open-source datasets. In addition, our data construction pipeline includes two verification stages to ensure that the localized cropped clip is sufficient to answer the question on its own and remains consistent with the original answer under full context. This provides a functional guarantee of grounding quality for retained samples.
>
> For the RL data, we further performed manual checking and correction, discarding or fixing incorrect samples so that each grounding-QA pair is strictly valid. According to our human annotation records, on videos longer than 3 minutes, the pipeline achieves 89.4% annotation accuracy, with 0.451 mIoU against human-labeled intervals, which strongly supports the quality of the generated data.
>
> We also tested Qwen3-VL-235B-A22B as the teacher:
>
> |Teacher Model|MLVU|VideoMMMU|VideoMME|LVBench|
> |-|-|-|-|-|
> |Qwen3-VL-235B-A22B|52.1|52.8|64.2|**43.2**|
> |Gemini2.5-Pro|**54.2**|**53.2**|**64.5**|43.0|
>
> These results suggest that strong open-source models such as Qwen3-VL are already capable of constructing high-quality thinking-with-videos data at lower cost.
>
>
> > W3 / Q3 / Limitation: Poor performance on videos >20min.
>
> The degradation is indeed related to the issue you raised. Due to context-length and computational constraints, the initial sampling for very long videos is inevitably sparse, so the model can observe only a limited number of frames. Even with the proposed thinking-with-videos paradigm, question-relevant visual evidence may still be missed. If a key event is completely missed in the initial stage, later refinement may not be able to recover it. In addition, compared with short videos, ultra-long videos contain much more redundant information, such as repeated scenes and transition segments, which further distracts the model from identifying critical evidence.
>
> To verify the importance of long-video training data, we removed all training samples longer than 5 minutes and obtained the following results:
>
> |Method|Short|Medium|Long|Overall|
> |-|-|-|-|-|
> |Qwen2.5-VL-7B|69.8|59.2|50.8|59.9|
> |VideoTemp-o3 w/o long video|**72.8**|62.5|51.2|62.1|
> |VideoTemp-o3|72.2|**66.6**|**54.7**|**64.5**|
>
> Removing long-video samples slightly improves performance on short (<3min) videos, but hurts performance on both medium (3-20min) and long (>20min) videos. This confirms that long-video training data is important for improving long-video understanding.
>
> > Concern 1: Key information about VideoTemp-Bench only in appendix.
> > Concern 2: Gemini-1.5-Pro and GPT-4o should be framed as "out-of-domain" baselines.
>
> Thank you for the suggestion. We will revise and emphasize these points in the revised version.
>
> > Q4: Have you considered integrating external tools (e.g., OCR models, object detectors) into the agentic pipeline?
>
> Thank you for the suggestion. Fine-grained tasks such as counting and OCR remain common weaknesses of current video models. Our framework adopts an agentic pipeline, which naturally supports the integration of external tools through the tool-call interface. In principle, the tool set can be extended to include OCR models, object detectors, spatial cropping/zooming tools, and others. We plan to explore such integrations in future work to further improve performance on complex video understanding tasks.

---

> > ### Author Rebuttal · Reviewer_nabb · 2026-04-02
> >
> > Thank you for the detailed rebuttal and additional experiments. My concerns are largely addressed, and I decide remain my score. I encourage the authors to incorporate these additional results into the future version to further strengthen the paper.

---

> > > ### Author Response · Authors · 2026-04-02
> > >
> > > Dear Reviewer nabb:
> > >
> > > Thank you for your thoughtful and comprehensive feedback. We will carefully incorporate your suggestions to further strengthen the manuscript.

---

### Official Review · Reviewer_fuyS · 2026-03-07

**Soundness:** 4
**Presentation:** 3
**Significance:** 3
**Originality:** 2
**Overall Recommendation:** 4
**Confidence:** 4

**Summary:**

This paper studies grounded long-video QA in the thinking-with-videos setting.

It proposes a unified grounding + QA model with on-demand cropping, curated multi-turn training data, and task-specific SFT/RL (notably unified masking and a penalty-aware IoU reward).

Experimental results on multiple datasets verify the effectiveness of the overall approach. The accompanying analyses also appear fairly comprehensive.

**Compliance With Llm Reviewing Policy:**

Affirmed.

**Final Justification:**

Overall, my major concerns are largely addressed. In particular, W4 is addressed well, W2 is partially addressed, and W3 is reasonably explained. For W2, while the added evidence on robustness and the use of open-source models are valuable, the pipeline still appears to require relatively large models and nontrivial cost. For W3, the heuristic is reasonably justified, but the original concern about task-/trajectory-specificity still partially remains.

That said, the rebuttal makes the method appear substantially more robust and generalizable than before, so I will revise my score accordingly. I encourage the authors to further investigate these aspects in future work and to more clearly explain the scope and limitations of the method in the paper.

**Key Questions For Authors:**

My overall impression is incremental but sound: the work is engineering-heavy and somewhat task-specific, but the execution is careful and the empirical package is reasonably solid within its chosen setting. That said, if the method is shown to be sufficiently robust and generalizable (especially with respect to W2-W4), it would still be a valuable contribution to the community. I would be open to revising my score if these key concerns are addressed and/or if any misunderstanding or oversight on my part is clarified.

**Limitations:**

I do not see a limitation section. The paper does touch on some remaining challenges through its analysis section and includes an impact statement, but it would benefit from a more explicit limitations discussion. In particular, I encourage the authors to discuss the oracle dependence of the data curation pipeline, the extent to which the masking and reward design generalize beyond this specific setting.

**Strengths And Weaknesses:**

## Strengths

1. The overall system design is coherent. Unifying grounding and QA in one model with optional on-demand cropping is cleaner than a multi-model pipeline. The data curation effort is also substantial.
2. The training choices are well motivated for this setup. In particular, unified masking and the penalty-aware IoU reward are supported by ablations suggesting that both matter in practice.
3. Experiments are fairly thorough: long-video understanding, temporal grounding, grounded QA, and duration-stratified analysis are all included.


## Weaknesses

1. Novelty is moderate. The overall recipe still lies within the localize-crop-answer / tool-use / SFT+RL family, and much of the gain appears to come from system integration, data engineering, and reward shaping rather than a major conceptual advance.
2. A large fraction of the pipeline is oracle-heavy. Strong teacher models and human cleanup play a central role in trajectory generation, relocalization, and verification, which raises concerns about cost, reproducibility, and transferability.
3. The unified masking rule looks fairly task-/trajectory-specific. Its success seems tied to the authors' assumption that only the last two turns are reliably correct; the paper does not show that this generalizes beyond this particular data construction scheme.
4. The RL reward design also seems carefully calibrated to this setting. The paper shows that it is better than a naive IoU reward here, but does not establish robustness across datasets, grounding granularities, or reward hyperparameters.
5. While the proposed components outperform the vanilla baselines, some important controls are missing. I would have liked to see stronger same-backbone baselines such as always-crop-once, separate grounder+QA, prompt-only re-grounding, or simple inference-time self-correction/reconsideration baselines.
6. Support for harder compositional cases is unclear. The method appears centered on a single contiguous span with limited refinement, so multi-span or A+B+C evidence gathering remains underexplored.
7. The limitation section seems to be missing if I have not overlooked.

---

> ### Author Rebuttal · Authors · 2026-03-30
>
> Thank you for your valuable feedback. We hope our responses address your concerns.
>
> > W1: Novelty appears moderate.
>
> We understand the concern that our framework may seem like an instance of the broader localize-crop-answer paradigm. However, we believe VideoTemp-o3 goes beyond system integration in two key aspects.
>
> First, it unifies temporal grounding and VideoQA in a single model. Unlike prior work, which only inserts timestamps in text-only CoT without tool calls, our method treats grounding as an explicit first-class task. This makes the model’s grounding ability directly measurable and optimizable. Our experiments further show that, especially for long videos, QA performance is highly correlated with grounding quality, highlighting the importance of intrinsic grounding ability.
>
> Second, we introduce a unified masking mechanism to address noisy supervision in multi-turn trajectory training. This is a general training strategy for multi-turn thinking-with-images/videos tasks. By masking earlier potentially noisy turns, the model can learn more stable localization and answering behaviors, while still retaining partial access to prior turns and thus preserving some exploratory behavior.
>
> > W2: The pipeline is oracle-heavy.
>
> Beyond Gemini-2.5-Pro, we also experimented with Qwen3-VL-235B-A22B as the teacher, which is open-source and substantially cheaper for data construction. The resulting model achieves:
>
> https://anonymous.4open.science/r/27F1/fuyS_W2.png
>
> These results show that strong open-source models, such as the Qwen3-VL series, are already capable of producing high-quality thinking-with-videos data, reducing construction cost while maintaining competitive downstream performance. We ultimately chose Gemini-2.5-Pro for the best overall quality.
>
> More broadly, we view the proposed pipeline itself as a contribution: it is a general and effective data construction framework with two-stage verification to ensure clip quality, and it can be used to build high-quality thinking-with-videos trajectory data.
>
> > W3: Unified masking may be trajectory-specific.
>
> In multi-turn thinking-with-images/videos, the reasoning trajectory often involves several attempts, with the penultimate turn typically making the tool call and the final turn producing the answer. In this setting, the last localization is usually the most reliable and directly relevant to the answer. Our unified masking is a simple yet effective strategy: it masks potentially noisy earlier turns so that the model can learn correct localization and answering more stably, while still preserving some exposure to prior context for exploration.
>
> We acknowledge that if future data construction methods can guarantee high quality for every turn, the need for masking may decrease. Under current practical conditions, however, we find it to be a simple and effective solution.
>
> > W4: Reward design seems highly calibrated.
>
> We would like to clarify the robustness of our reward design. The proposed reward consistently improves performance across multiple benchmarks, none of which overlap with the training data, suggesting good generalization:
>
> https://anonymous.4open.science/r/27F1/fuyS_W4_1.png
>
> We also evaluate localization quality (mIoU) under different grounding granularities using part of ScaleLong:
>
> https://anonymous.4open.science/r/27F1/fuyS_W4_2.png
>
> These results indicate that the proposed penalty-aware IoU reward consistently improves mIoU across different localization granularities. Due to rebuttal length constraints, we refer the reviewer to our response to `Reviewer nabb, Q1` for the reward hyperparameter.
>
> > W5: Missing stronger same-backbone baselines.
>
> Due to time and resource constraints during rebuttal, we did not train all these baselines separately. Instead, we simulated them through prompting at evaluation time: for VideoTemp-o3, we enforced crop-once and direct answer settings; for Qwen, we tested prompt-only re-grounding and inference-time self-correction.
>
> https://anonymous.4open.science/r/27F1/fuyS_W5.png
>
> VideoTemp-o3 outperforms all these baselines. We will include them in the revised paper for more direct comparison.
>
> > W6: Multi-span evidence gathering is underexplored.
>
> We have already considered A-or-B-or-C style multi-span evidence; please see our response to `Reviewer shMG, Q4`.
>
> For A+B+C compositional evidence gathering in long-video understanding, we agree this is not yet fully explored. This is partly because of current model limitations and most existing benchmarks can be answered with one evidence segments. We will discuss this more explicitly in future work, including extensions such as allowing tool calls to output multiple temporal spans per turn or designing new rewards that encourage joint localization of multiple relevant clips.
>
> > W7 / Limitation: Missing limitation section.
>
> We apologize for the omission. In the revised paper, we will include a full Limitations section addressing your concerns.

---

> > ### Author Rebuttal · Reviewer_fuyS · 2026-04-03
> >
> > Overall, my major concerns are largely addressed. In particular, W4 is addressed well, W2 is partially addressed, and W3 is reasonably explained. For W2, while the added evidence on robustness and the use of open-source models are valuable, the pipeline still appears to require relatively large models and nontrivial cost. For W3, the heuristic is reasonably justified, but the original concern about task-/trajectory-specificity still partially remains.
> >
> > That said, the rebuttal makes the method appear substantially more robust and generalizable than before, so I will revise my score accordingly. I encourage the authors to further investigate these aspects in future work and to more clearly explain the scope and limitations of the method in the paper.

---

> > > ### Author Response · Authors · 2026-04-03
> > >
> > > Dear Reviewer fuyS,
> > >
> > > Thank you for your positive feedback and for recognizing the value of our work. We are very pleased to know that your concerns have been largely addressed, and we will further improve the paper based on your suggestions.

---

### Official Review · Reviewer_shMG · 2026-03-09

**Soundness:** 3
**Presentation:** 2
**Significance:** 2
**Originality:** 2
**Overall Recommendation:** 4
**Confidence:** 4

**Summary:**

This paper proposes VideoTemp-o3, a unified framework for thinking-with-videos. The model combines temporal grounding and video question answering in one system. It follows a localize-clip-answer process. The authors design a supervised fine tuning strategy with a unified masking mechanism to reduce noise from incorrect early grounding steps. They also introduce reinforcement learning with accuracy, format, and penalty-aware IoU rewards to improve grounding quality and avoid reward hacking. In addition, the paper builds a data pipeline for multi-turn grounded video QA training and proposes VideoTemp Bench to evaluate performance on videos of different lengths. Experiments show that the method improves results on several video understanding and temporal grounding benchmarks.

**Compliance With Llm Reviewing Policy:**

Affirmed.

**Final Justification:**

The author has addressed most of the detailed issues I raised in the review, and I revise my original score of 3 (weak reject) to 4 (weak accept). Recently, there have been a large number of papers on thinking with videos and temporal grounding, including five works I mentioned in my review (e.g., VideoChat-R1.5) as well as concurrent studies in the past five months (such as Conan, VideoZoomer, etc.), which share similarities in the final functional implementation. I hope the authors can provide a more in-depth discussion in the revised version, covering the advantages of the reasoning paradigm, efficiency, and a fair comparison of performance.

**Key Questions For Authors:**

1. The paper does not compare with several recent works (before 2025.10.31), such as VideoChat-R1.5, VideoThinker, Open-o3 Video, Rewatch-R1, and FrameThinker. Can the authors clarify the main differences between this work and these methods and explain the main novelty?

2. The paper proposes VideoTemp-Bench. What is the main difference between this benchmark and existing benchmarks such as CG-Bench? What new insights does it provide for evaluating video reasoning models? Is the benchmark data manually verified? How do the authors ensure the quality and correctness of the annotations?

3. Some experimental settings are unclear. For example, does the MLVU evaluation use the dev set or the test set? In multi-turn dialogue, how are frames sampled from the cropped clips in later turns? The number of dialogue turns is also not clearly described. During training and inference, what is the average number of turns and what is the maximum allowed number of turns?

4. For temporal grounding annotations, is it possible that the evidence used to answer a question is outside the labeled time segment but still allows the question to be answered (one question but multi-evidence) ? If this happens, how is this situation handled to avoid evaluation errors?

5. The reported results of Qwen2.5-VL-7B differ from results reported in other papers, such as official reports and VideoZoomer results on VideoMMMU, with gaps larger than about 4%. In addition, your setting seems to use more input frames. Can the authors clarify the exact evaluation setting and explain the reason for this difference? Is the reported result obtained with a think then answer process (use cot) ? If so, what is the performance when the model directly outputs the answer without the thinking stage? Does the thinking with videos reasoning process improve performance compared with direct answering?

6. The experiments use Qwen2.5-VL-7B as the baseline. Would this training paradigm and dataset still be effective on stronger recent models such as Qwen3-VL-4B/8B?

**Limitations:**

No. The authors should explain the differences from related work more clearly and provide more complete experimental details (see questions) . They can also discuss more about: what types of videos or questions the current method still cannot handle well (maybe some bad cases analysis), and what directions may improve the framework in future work.

**Strengths And Weaknesses:**

Strengths

1. The data curation pipeline and the training pipeline with SFT and RL are technically sound. The paper also includes new designs to reduce reward hacking and reasoning noise.

2. The paper studies the thinking with videos paradigm, which is an important direction for video MLLMs.

3. The experiments are extensive, and the analysis is relatively thorough.


Weaknesses

1. Some experimental settings and details are unclear. See questions.

2. The paper does not compare with some recent thinking-with-videos related works (see questions) , which makes the novelty less clear.

3. The paper introduces a new benchmark. However, the difference from existing benchmarks such as CG-Bench is not clearly explained.

---

> ### Author Rebuttal · Authors · 2026-03-30
>
> We sincerely thank the reviewer for taking the time to evaluate our paper and for providing valuable feedback. We hope our responses below adequately address your concerns.
>
> > Q1: Missing comparisons and novelty vs. recent methods?
>
> Thank you for pointing out these important related works.
> - VideoChat-R1.5, VideoThinker, Open-o3 Video, and Rewatch-R1 can output timestamps or bounding boxes during reasoning, but they all follow a one-shot text-only CoT paradigm without tool calls, and therefore cannot support multi-round localization and answering.
> - FrameThinker enables multi-turn visual interaction, but it does not unify localization and QA in a single framework, nor does it introduce dedicated designs to mitigate reward hacking.
>
> In contrast, VideoTemp-o3 unifies temporal grounding and VideoQA in one model, includes a refinement mechanism for initial localization, and supports an on-demand, iterative localize-crop-answer pipeline. The model can adaptively decide whether cropping and refinement are necessary based on video complexity.
>
> > Q2: Difference between VideoTemp-Bench and CG-Bench?
>
> The two benchmarks target different goals. VideoTemp-Bench focuses on Grounded VQA, evaluating both temporal grounding and QA across videos of different lengths, with mIoU and Acc as the main metrics. In contrast, CG-Bench emphasizes evidence retrieval and answer faithfulness in long videos, using CRR and Acc as the primary metric.
>
> In addition, VideoTemp-Bench covers a broader spectrum of video lengths (0–3 min, 3–10 min, 10–20 min, >20 min), enabling systematic analysis across short to ultra-long videos, while CG-Bench only considers videos longer than 10 minutes. VideoTemp-Bench is built by adapting open-source datasets and uses a two-stage verification process to ensure data quality. For RL data construction, we further perform manual validation and correction/discarding of noisy samples so that each grounding-QA pair is strictly valid.
>
> > Q3: Experimental settings are unclear.
>
> We use the MLVU test set, which contains more diverse question types and is more challenging.
>
> For each tool-call crop, the video segment is sampled at 2 FPS, with at most 64 frames, at a resolution of 224×224. Due to resource constraints, the maximum number of tool calls is 3 during both training and inference, which effectively allows at most one round of timestamp refinement.
>
> As shown in Fig. 7(b), on VideoMME / VideoTemp-Bench, the average number of tool calls for short (0–3 min), medium (3–20 min), and long (>20 min) videos is 0.47/0.32, 1.46/1.30, and 1.60/1.60, respectively, indicating that the model adaptively adjusts the number of calls based on video length.
>
>
> > Q4: Multi-evidence questions: could valid evidence lie outside the labeled segment?
>
> Yes, this issue is important, and we have explicitly considered it. For RL data, we manually annotate all time intervals relevant to the question, so that the model is not wrongly penalized for localizing an alternative valid segment. This encourages exploration of all possible evidence spans.
>
> For SFT data, we use a two-stage verification pipeline to ensure that the localized segment is sufficient to answer the question, so each grounding-QA pair is valid even if other relevant segments also exist.
>
> > Q5: Qwen2.5-VL-7B results differ from official reports. CoT vs. direct answering?
>
> For fairness, we reproduced Qwen2.5-VL-7B ourselves. Compared with the official setting (about 776×776, max 768 frames), we use a lower resolution but more frames (224×224, 1024 frames) so that the model can skim more coarse information from long videos before localization, which better fits our setting. During evaluation, Qwen answers directly without CoT.
>
> We also tested a higher-resolution setting:
>
> https://anonymous.4open.science/r/27F1/shMG_Q5.png
>
> These results show that resolution has a strong impact on performance, and VideoTemp-o3 can better leverage denser video coverage for localization and answering. Interestingly, for Qwen, direct answering performs better than CoT in our setting, possibly because it was not specifically trained for video reasoning.
>
> > Q6: Would this training paradigm work on stronger models such as Qwen3-VL-4B/8B?
>
> We chose Qwen2.5-VL-7B because most related work uses it as the backbone, allowing fairer comparison. Due to rebuttal-time constraints, we conducted a small-scale RL experiment on Qwen3-VL-4B, using about 900 samples for 100 steps:
>
> https://anonymous.4open.science/r/27F1/shMG_Q6.png
>
> The gains indicate that our data and training paradigm can also help stronger backbones learn a thinking-with-videos behavior, consistent with what we observed on Qwen2.5-VL. We leave larger-scale experiments on Qwen3-VL to future work.
>
> > Limitations
>
> We appreciate the suggestion and will add a fuller discussion of limitations and future work in the revised paper, including failure cases on certain video/question types and possible directions to improve the framework.

---

> > ### Author Rebuttal · Reviewer_shMG · 2026-04-03
> >
> > Thank you to the authors for the effort put into the rebuttal. It has addressed most of my concerns.
> >
> > **Follow-up questions:**
> >
> > **(1) About Q1:**
> > I still have some reservations about the explanation of VideoChat-R1.5. Specifically, the work presented at NeurIPS 2025 ([https://neurips.cc/virtual/2025/loc/san-diego/poster/116032](https://neurips.cc/virtual/2025/loc/san-diego/poster/116032)) incorporates multi-turn tool usage (temporal zoom-in for video) rather than relying solely on pure text-based reasoning.
> >
> > This work also evaluates video understanding/reasoning and temporal grounding. In my view, it would strengthen the paper to provide a clearer comparison with this line of work, both in terms of novelty and experimental performance.
> >
> > **(2) About Q5:**
> > In my original question, I referred to discrepancies in the results on VideoMMMU. However, the figure provided in the rebuttal appears to report results on VideoMME instead. I would appreciate clarification on this point.
> >
> >
> > If these issues are adequately addressed, I would be willing to consider increasing my score.

---

> > > ### Author Response · Authors · 2026-04-03
> > >
> > > Dear Reviewer shMG:
> > >
> > > Thank you for your feedback. We are glad to see that most of your concerns have been addressed, and we apologize for the parts that were unclear in our earlier rebuttal. We hope to further address your remaining questions.
> > >
> > > > (1) Comparison with VideoChat-R1.5 and tool-based video reasoning work: novelty and performance differences?
> > >
> > > Thank you for pointing this out, and we apologize that our previous explanation of VideoChat-R1.5 was not sufficiently clear.
> > >
> > > That said, we still consider VideoChat-R1.5 to follow a **text-only CoT paradigm without tool calls**. Beyond the paper itself, we carefully examined its implementation code and found that its video reasoning process works as follows:
> > >
> > > 1. The model first reasons over the video and question, and outputs the reasoning process, answer, and timestamps of relevant video segments.
> > > 2. The model then reasons over the video and the question again. In this step, frames within the previously predicted key timestamp intervals are sampled more densely, while frames outside those intervals are sampled normally. This allows the model to access more potentially useful video information. It again outputs the reasoning process, the answer, and timestamps of the video segments to attend to.
> > > 3. Step 2 is repeated until the predefined maximum number of perception rounds is reached.
> > >
> > > Therefore, in each round, VideoChat-R1.5 takes a single video input, possibly with denser frame sampling over selected intervals, together with the text question, and produces a purely textual CoT and answer. In our view, this differs from VideoTemp-o3, which is a multi-round, multimodal CoT framework with tool calls. Moreover, although VideoChat-R1.5 can repeatedly localize relevant segments, each reasoning round remains independent, and timestamps are produced only when explicitly requested by the prompt. Therefore, we do not consider it capable of on-demand tool call and iterative refinement of previous localization results in the same way as VideoTemp-o3.
> > >
> > > As you pointed out, VideoChat-R1.5 is indeed evaluated on both grounding and QA tasks, and is therefore worth comparing against. However, we note that the maximum video resolution and number of frames used in VideoChat-R1.5 are aligned with the official Qwen2.5-VL setting, i.e., 768×28×28 (approximately 776×776) and 768 frames, whereas VideoTemp-o3 uses 64×28×28 (224×224) and 1024 frames. As shown by our experiments in `Q5` and the `next response`, resolution has a substantial impact on evaluation results. Therefore, a direct comparison between the numbers reported in the two papers would not be fair.
> > >
> > > To enable a fair comparison, we evaluated VideoChat-R1.5 under the same resolution and frame settings as VideoTemp-o3 on a subset of grounding and QA benchmarks:
> > >
> > > |Charades-STA|R\@0.7|mIoU|
> > > |-|-|-|
> > > |VideoChat-R1.5|20.2|38.0|
> > > |VideoTemp-o3|**33.0**|**57.8**|
> > >
> > > |VideoMMMU|Adapt.|Compr.|Prec.|Avg.|
> > > |-|-|-|-|-|
> > > |VideoChat-R1.5|40.6|45.1|63.2|49.6|
> > > |VideoTemp-o3|**43.0**|**47.8**|**69.0**|**53.2**|
> > >
> > > |VideoMME|Short|Medium|Long|Overall|
> > > |-|-|-|-|-|
> > > |VideoChat-R1.5|**72.3**|63.9|54.1|63.4|
> > > |VideoTemp-o3|72.2|**66.6**|**54.7**|**64.5**|
> > >
> > > As shown above, under the same evaluation conditions, VideoTemp-o3 outperforms VideoChat-R1.5 on grounding and QA benchmarks overall, demonstrating stronger grounding and video understanding capabilities.
> > >
> > > In the revised version of the paper, we will further expand the discussion of related work, including VideoChat-R1.5, and add fairer controlled comparisons wherever possible.
> > >
> > >
> > > > (2) The question concerns VideoMMMU, but the rebuttal reports results on VideoMME.
> > >
> > > Thank you for pointing this out. We apologize for mistakenly reporting results on VideoMME in our previous rebuttal. We have now further evaluated on VideoMMMU under the same settings:
> > >
> > > |VideoMMMU|Adapt.|Compr.|Prec.|Avg.|
> > > |-|-|-|-|-|
> > > |Qwen-Offical (776\*776, 768)|-|-|-|47.4|
> > > |Qwen (448\*448, 768)|36.0|46.0|58.2|46.7|
> > > |Qwen-CoT (448\*448, 768)|36.0|43.2|53.6|44.2|
> > > |VideoTemp-o3 (448\*448, 768)|38.6|46.2|60.6|48.5|
> > > |Qwen (224\*224, 1024)|35.9|36.1|57.6|43.2|
> > > |VideoTemp-o3 (224\*224, 1024)|**43.0**|**47.8**|**69.0**|**53.2**|
> > >
> > > Consistent with the VideoMME results discussed in `Q5`, increasing the resolution from 224×224 to 448×448 further improves the performance of Qwen2.5-VL and brings it closer to the official result, highlighting the importance of resolution for Qwen. At the same time, under our original paper setting (224×224, 1024), VideoTemp-o3 still achieves the best performance, validating our design choice of lowering the resolution to increase the frame rate, so that the model can skim more frames and better support thinking with videos.
> > >
> > > Thank you again for the helpful feedback and hope this clarification addresses your remaining concerns.

---

### Official Review · Reviewer_Z8cv · 2026-03-13

**Soundness:** 4
**Presentation:** 3
**Significance:** 3
**Originality:** 2
**Overall Recommendation:** 5
**Confidence:** 4

**Summary:**

The paper presents a unified framework that uses a single model to perform both temporal grounding and question answering. The model first uses a coarse sampling of the entire video to roughly localize events of interests, crops the video to the segment, repeats these steps until an answer is evident, and then answers the question. They implement a data generation pipeline to facilitate SFT and then propose a reward structure for a GRPO RL pipeline to self-improve model capabilities. They find SOTA performance on final answer metrics, and competitive performance on the intermediate task of temporal grounding.

**Compliance With Llm Reviewing Policy:**

Affirmed.

**Final Justification:**

Pls see my comment in the rebuttal acknowldgement.  I think the paper advances the spaces (or at least delves into a very interesting area within video understanding and RL).  The method (like all methods) has limitations; the paper would be stronger if these were better discussed and acknowledged.

**Key Questions For Authors:**

Is the reason that long videos (>20 minutes) still present a challenge due to the initial sparse sampling of frames missing key events?

If there are events in a video that last for only a few frames, is it probable that these events will be missed even by this unified model? Are other mechanisms needed to search videos for such events?

Given that there is a reward term that depends on having ground truth intervals, can this method self-improve given data that consists only of (video, question, answer) tuples? Or is it restricted to datasets that consist of (video, question, temporal interval, answer) tuples?

**Limitations:**

The paper has a dismissive statement about societal limitations.

>> revised post rebuttal
if accepted, the paper needs to be much clearer about its limitations (social and technical), these partially were confirmed in the rebuttal.  (I nevertheless think the paper should be accepted).

**Strengths And Weaknesses:**

Strengths

The generall narrative is easy to follow. Figures explain the method well.

This paper introduces a novel masking strategy for generating multi-turn SFT data that is meaningfully different than prior works like LongVT and may contribute to the increased performance.

The paper compares the learned model to the SOTA using empirical results on known long video comprehension datasets as well as on the benchmark generated for this paper. These cover a large range of video lengths and provides support for the claim that this model is now SOTA in long video question answering for videos up to 20 minutes.

Long form video QA is an important task with broad applications. Methods developed for this task also translate well to other long sequence understanding tasks. This paper will likely influence future works in the methodology of producing SFT training samples and rewards for RL fine tuning.

The paper follows the strategy of prior works like LongVT by using a chain of thought strategy with repeated global-local reductions to search videos for context and extract relevant information from fine-grained crops.

Weaknesses

The unified framework used in this paper creates a sequential processing bottleneck where multiple cropped video clips cannot be analyzed in parallel. This could increase computation time significantly, but performance metrics are not provided so it is difficult to know.

The paper lacks any discussion of real hypotheses about why the stated limitations exist, limiting the possible analysis by the reader.

---

> ### Author Rebuttal · Authors · 2026-03-30
>
> We sincerely appreciate the reviewer for the constructive comments and the supportive assessment of our paper. We hope the responses below address your points effectively.
>
> > W1: Sequential bottleneck; no efficiency analysis.
>
> We agree that the unified framework introduces some inference overhead. However, this design is necessary to support the iterative refinement mechanism. In our practical tests, each video cropping operation takes about 0.7s. Since the model can decide on demand whether further localization is needed, the efficiency gap on short videos is small. For medium and long videos, although some additional overhead exists, we believe it is acceptable given the performance gains.
>
> On VideoMME, the average number of tool calls is only 0.47 for short videos, 1.46 for medium videos, and 1.60 for long videos. This suggests that the model does not blindly perform multiple rounds of cropping for every video, but instead invokes refinement only when necessary.
>
> > W2 / Limitation: Lack of discussion on limitations and hypotheses.
>
> Thank you for the suggestion. In the revision, we will substantially expand the Limitations section. In particular, we will discuss the performance bottleneck on very long videos, the reliance of our data construction pipeline on a strong teacher model, and the limitations related to localization rounds and the choice of base model. We will also provide more concrete hypotheses for why these limitations arise and outline possible future directions.
>
> > Q1: Are long videos challenging because sparse initial sampling misses key events?
>
> Yes. Due to context-length and computational constraints, videos longer than 20 minutes, especially hour-long ones, must be sampled very sparsely at the initial stage. As a result, the model can observe only a limited number of frames, making it easy to miss question-relevant visual evidence. If key events are missed entirely, later refinement may not be able to recover from this error. In addition, very long videos contain substantial redundant information, such as repeated scenes and transition segments, which can further distract the model from identifying critical evidence.
>
> That said, VideoTemp-o3 still outperforms baseline models on very long videos—for example, compared with the second-best baseline VideoChat-R1, it achieves +2.2 mIoU and +1.0 Acc—showing that it partially alleviates this issue. In future work, we plan to explore hierarchical sampling and the use of video semantic structure (e.g., scene boundary detection) to improve initial sampling.
>
>
> > Q2: Can very short events still be missed?
>
> Yes, this remains a challenging problem and deserves further study. Compared with prior methods that perform one-shot reasoning over sparse samples, the iterative grounding mechanism in VideoTemp-o3 gives the model a second chance to localize key evidence. Even if a brief event is missed during initial sampling, the model may still infer a plausible temporal region from contextual cues and then issue a tool call to crop that segment at a higher sampling rate. This improves the chance of capturing fine-grained details. Nevertheless, more specialized mechanisms may still be needed. Future improvements may include finer temporal modeling and additional localization rounds to better detect extremely short events.
>
> > Q3: Can the method self-improve with only (video, question, answer) data?
>
> Our RL training can still support self-improvement with only (video, question, answer) tuples, but the gains are smaller without the direct localization supervision provided by the IoU reward:
>
> |Method|MLVU|VideoMMMU|VideoMME|LVBench|
> |-|-|-|-|-|
> |Qwen-2.5-VL-7B|45.2|43.2|59.9|39.2|
> |+SFT|49.5|48.7|60.6|39.6|
> |+RL (w/o IoU)|50.6|51.6|63.3|41.7|
> |+RL|**54.2**|**53.2**|**64.5**|**43.0**|
>
> Although removing the IoU reward leads to lower performance, RL without IoU still outperforms SFT, indicating that the accuracy reward and format reward can indirectly encourage better localization behavior. However, direct localization supervision is still very important for learning an effective localize-crop-answer process, and adding this reward further improves overall performance.

---

> > ### Author Rebuttal · Reviewer_Z8cv · 2026-04-02
> >
> > My concerns are largely addressed in the rebuttal, thank the authors for that.  In fact, they are addressed in two capacities: (1)  certain limitations were answered and with data to resolve the concerns, (2) certain questions were answered that confirmed limitations of the work.  I think the work is timely and reasonable.  A worthy paper to accept.  BUT I think the paper needs to be more clear about its limitations.

---

> > > ### Author Response · Authors · 2026-04-02
> > >
> > > Dear Reviewer Z8cv:
> > >
> > > Thank you for your thoughtful and encouraging feedback. We are glad that our rebuttal addressed most of your concerns, and we appreciate your positive assessment of the paper. We will revise the final version based on your suggestions to make the limitations of our work more explicit and clear.

---

### Decision · Program_Chairs · 2026-04-30

**Decision:**

Accept (regular)

**Comment:**

This paper proposes VideoTemp‑o3, a unified agentic framework that jointly models temporal grounding and video question answering for long‑video understanding. Reviewers agreed that the proposed approach is technically sound, empirically well‑validated, and addresses an important and emerging direction in multimodal reasoning.
The author rebuttal was constructive and materially improved reviewer confidence. It clarified experimental settings, provided sensitivity analysis for the reward design, added controlled comparisons with recent tool‑based reasoning methods (e.g., VideoChat‑R1.5), and offered additional evidence regarding annotation quality and robustness across benchmarks. As a result, reviewers either raised their score or keep their positive ratings. The final scores are 5 / 4 / 4 / 4.
Overall, the committee finds that the paper makes a timely and useful contribution to the growing literature on agentic video reasoning and long‑sequence multimodal understanding. The AC agrees with the consensus and recommend acceptance.